# Comparison of intramyocellular lipid metabolism in patients with diabetes and male athletes

Alice M. Mezincescu[1,9], Amelia Rudd[1,9], Lesley Cheyne[1], Graham Horgan [2], Sam Philip [1], Donnie Cameron[3], Luc van Loon[4], Phil Whitfield[5], Rachael Gribbin[6], May Khei Hu[1], Mirela Delibegovic [1], Barbara Fielding[6], Gerald Lobley[1], Frank Thies[1], David E. Newby [7], Stuart Gray [5], Anke Henning[8] & Dana Dawson [1] ✉

Despite opposing insulin sensitivity and cardiometabolic risk, both athletes and patients with type 2 diabetes have increased skeletal myocyte fat storage: the so-called "athlete's paradox". In a parallel non-randomised, non-blinded trial (NCT03065140), we characterised and compared the skeletal myocyte lipid signature of 29 male endurance athletes and 30 patients with diabetes after undergoing deconditioning or endurance training respectively. The primary outcomes were to assess intramyocellular lipid storage of the vastus lateralis in both cohorts and the secondary outcomes were to examine saturated and unsaturated intramyocellular lipid pool turnover. We show that athletes have higher intramyocellular fat saturation with very high palmitate kinetics, which is attenuated by deconditioning. In contrast, type 2 diabetes patients have higher unsaturated intramyocellular fat and blunted palmitate and linoleate kinetics but after endurance training, all were realigned with those of deconditioned athletes. Improved basal insulin sensitivity was further associated with better serum cholesterol/triglycerides, glycaemic control, physical performance, enhanced post insulin receptor pathway signalling and metabolic sensing. We conclude that insulin-resistant, maladapted intramyocellular lipid storage and turnover in patients with type 2 diabetes show reversibility after endurance training through increased contributions of the saturated intramyocellular fatty acid pools. Clinical Trial Registration: NCT03065140: Muscle Fat Compartments and Turnover as Determinant of Insulin Sensitivity (MISTY)

Higher caloric intake and sedentary lifestyle lead to excess fat deposition during the long sub-clinical phase that precedes clinically overt type 2 diabetes and is the main contributor to its high morbidity and mortality[1]. Aside from increased subcutaneous and visceral adiposity, patients with type 2 diabetes also exhibit ectopic fat deposition within their skeletal myocytes[2–4]. This ectopic intramyocellular fat accumulation is intimately linked to the development of insulin resistance, which is the single most defining pathophysiological

[1]Aberdeen Cardiovascular and Diabetes Centre, University of Aberdeen, Aberdeen, UK. [2]Biomathematics & Statistics Scotland, Aberdeen, UK. [3]C.J. Gorter MRI Center, Leiden University Medical Center, Leiden, The Netherlands. [4]University of Maastricht, Maastricht, The Netherlands. [5]University of Glasgow, Glasgow, UK. [6]University of Surrey, Guildford, UK. [7]Centre for Cardiovascular Science, University of Edinburgh, Edinburgh, UK. [8]Southwestern University, Georgetown, TX, USA. [9]These authors contributed equally: Alice M. Mezincescu, Amelia Rudd. ✉e-mail: dana.dawson@abdn.ac.uk

characteristic of type 2 diabetes[5]. Surprisingly, increased intramyocellular fat storage also occurs in the skeletal muscle of healthy, endurance-trained athletes who, in contrast, are highly insulin sensitive[6]. The intramyocellular fat storage in athletes cannot be viewed as 'ectopic deposition' since athletes have very little subcutaneous adipose tissue, and this phenomenon is known as the "athlete's paradox"[7,8]. A mechanistic explanation for the increased intramyocellular lipid storage in participants who are at opposite ends of insulin sensitivity and cardiometabolic risk remains elusive and a matter of debate.

Important differences in intramyocellular lipids are likely to exist between insulin-sensitive athletes and insulin-resistant patients with type 2 diabetes, and if so, questions arise whether these differences could potentially be a target for manipulation in order to provide health improvements for patients. The abundance of specific intramyocellular lipid isoforms (diacylglycerols and sphingolipids) and their sub-cellular localisation have been proposed as potential candidates to subtend such differences and play a role in promoting insulin resistance[9,10]. However, these species represent only a small fraction of the intramyocellular lipid pool, which is largely composed of triacylglycerols. This suggests that further characteristics common to the majority of lipids must be present to explain the opposing phenotypes resulting from an apparently similar intramyocellular metabolic storage. The relative proportions or utilisation of saturated and unsaturated intramyocellular fat has been proposed as one such characteristic but has never been demonstrated. There is evidence from in vitro data that saturated and unsaturated fatty acids partition toward different metabolic pathways in muscle cells[11] and that upregulation of unsaturated lipo-synthetic pathways in human skeletal myocytes is strongly and reversibly associated with insulin resistance[12]. In keeping with this, a saturated high-fat diet increased content and activity of the mitochondrial fatty acid oxidation master regulator sirtuin in human skeletal muscle, whereas unsaturated diets did not[13]. Contrasting with these observations, saturated fatty acids are traditionally associated with increased cardio-metabolic risk. This creates equipoise regarding the relative contributions of saturated and unsaturated intramyocellular fat to insulin sensitivity.

To address this, we designed a clinical trial of age and sex-matched endurance athletes and patients with type 2 diabetes. The intervention was deconditioning for athletes and endurance exercise training for patients with type 2 diabetes, which influenced participants' basal insulin sensitivity status. At the distinct baseline and post-intervention stages, we captured the skeletal muscle spectroscopic detection of saturated versus unsaturated carbon bonds contained within intramyocellular lipid stores, stable isotope [U-13C] turnover of skeletal muscle saturated and unsaturated fatty acids, lipidomic, metabolic, molecular and physical performance profiling. We characterised the skeletal muscle intramyocellular lipid signature of athletic health *versus* diabetes dysmetabolism. The hypotheses of the study were: (1) the relative proportions of saturated/unsaturated intramyocellular lipid storage in patients with type 2 diabetes is different to age/sex matched healthy athletes, and (2) exercise endurance training in type 2 diabetes patients is associated with changes in the saturated/unsaturated intramyocellular storage and turnover.

## Results

### Participants

From 83 volunteers invited (39 endurance athletes and 44 patients with type 2 diabetes), 29 athletes and 30 patients with type 2 diabetes were enroled after screening (Fig. 1) between September 2016 and January 2019. One athlete and one patient withdrew, and two further patients were withdrawn after developing chest pain during the exercise training programme. Athletes were free of any health conditions or medications, whilst a third of patients were on antihypertensive medication and two thirds were on statin therapy for primary

prevention (Supplemental Table 1). At screening, athletes performed more daily exercise, but there were no differences in the dietary composition between groups for total fat, saturated, mono-unsaturated or poly-unsaturated fatty acids although athletes consumed more sugar. All participants had normal resting electrocardiograms and echocardiography-derived left ventricular ejection fraction, although, as expected, global left ventricular longitudinal strain was reduced in patients with diabetes and left ventricular volumes were increased in athletes, consistent with the "athletic heart". Contrast-enhanced cardiac magnetic resonance ruled out the possibility of any previously unknown (silent) myocardial infarction in patients with type 2 diabetes[14]. All participants achieved a respiratory exchange ratio >1.1 on cardiopulmonary exercise testing, and athletes exercised for longer times, achieved more metabolic equivalents and had higher peak oxygen consumption compared to patients with type 2 diabetes (Supplemental Table 1).

### Baseline characteristics of athletes and patients with type 2 diabetes

Athletes were leaner than patients (Table 1). First, we established the intramyocellular lipid storage pattern and Fig. 2A shows the schematic representation of the 1H-magnetic resonance spectroscopy acquisition of total, as well as saturated and unsaturated intramyocellular lipid peaks described in the methods. The fractional lipid mass (fLM) was lower in athletes' skeletal muscle compared to patients with type 2 diabetes ($0.035 \pm 0.01$ vs $0.056 \pm 0.01$ arbitrary units, $p < 0.0001$). In addition, athletes had higher fraction of intramyocellular saturated lipid ($87.96 \pm 4$ vs $81.65 \pm 6\%$, $p = 0.0004$) and lower fraction of intramyocellular unsaturated lipids ($12.04 \pm 4$ vs $18.35 \pm 6\%$, $p = 0.0004$) compared to patients with diabetes, Table 2 and Fig. 2B. Next, we assessed the skeletal muscle saturated and unsaturated fatty acid turnover, which is represented by fractional incorporation rate of each, calculated after intravenous administration of [U-13C]-palmitate (16.0) and [U-13C]-linoleate (18.2 n-6) coupled with pre- and post-infusion skeletal muscle biopsies. Both palmitate and linoleate fractional incorporation rates were higher in athletes compared to patients with diabetes ($0.62 \pm 0.4$ vs $0.15 \pm 0.07\%/h$ and $0.27 \pm 0.1$ vs $0.12 \pm 0.06\%/h$ respectively, a 4.1-fold for palmitate and 2.3-fold for linoleate) despite similar plasma rates of appearance, Table 3 and Fig. 3A. When the collective intra- plus extra-myocellular triacylglycerols, diacylglycerols and ceramide species were quantified directly from skeletal muscle biopsies and grouped according to saturation status, there were no differences between athletes and patients with type 2 diabetes in total, saturated or unsaturated lipid pools, Table 4. Western blot analyses of proteins involved in insulin receptor pathways also demonstrated no differences between athletes and patients with type 2 diabetes including the trans-membrane glucose transporters (GLUT1 and GLUT4), phosphorylated insulin receptor (pIR), insulin receptor substrate (IRS), phosphorylated S6 kinase protein (pS6), phosphorylated extracellular signal-regulated kinase (pERK), phosphorylated or total protein kinase B, phosphorylated or total 5' adenosine monophosphate-activated protein kinase or their phosphorylated to total ratios, all assessed in basal state (Table 4). Athletes had higher serum HDL-cholesterol concentrations but lower serum triglycerides, fasting glucose, insulin and plasma non-esterified (free) fatty acids and were highly insulin sensitive whereas patients with type 2 diabetes were insulin resistant (Table 5). Athletes' cardiopulmonary exercise performance was superior (Table 1) as already demonstrated at the screening stage (Supplemental Table 1). The forced expiratory volume in one second (FEV1), resting and peak oxygen saturations as well as the VE/VCO2 slope were within normal limits, ruling out any pulmonary pathology or cardiac limitation in either group.

Exercise interventions were group-specific: Compliance with deconditioning (from an average of 92 min to <10 min structured daily exercise for 4 weeks) by the athletes was monitored using

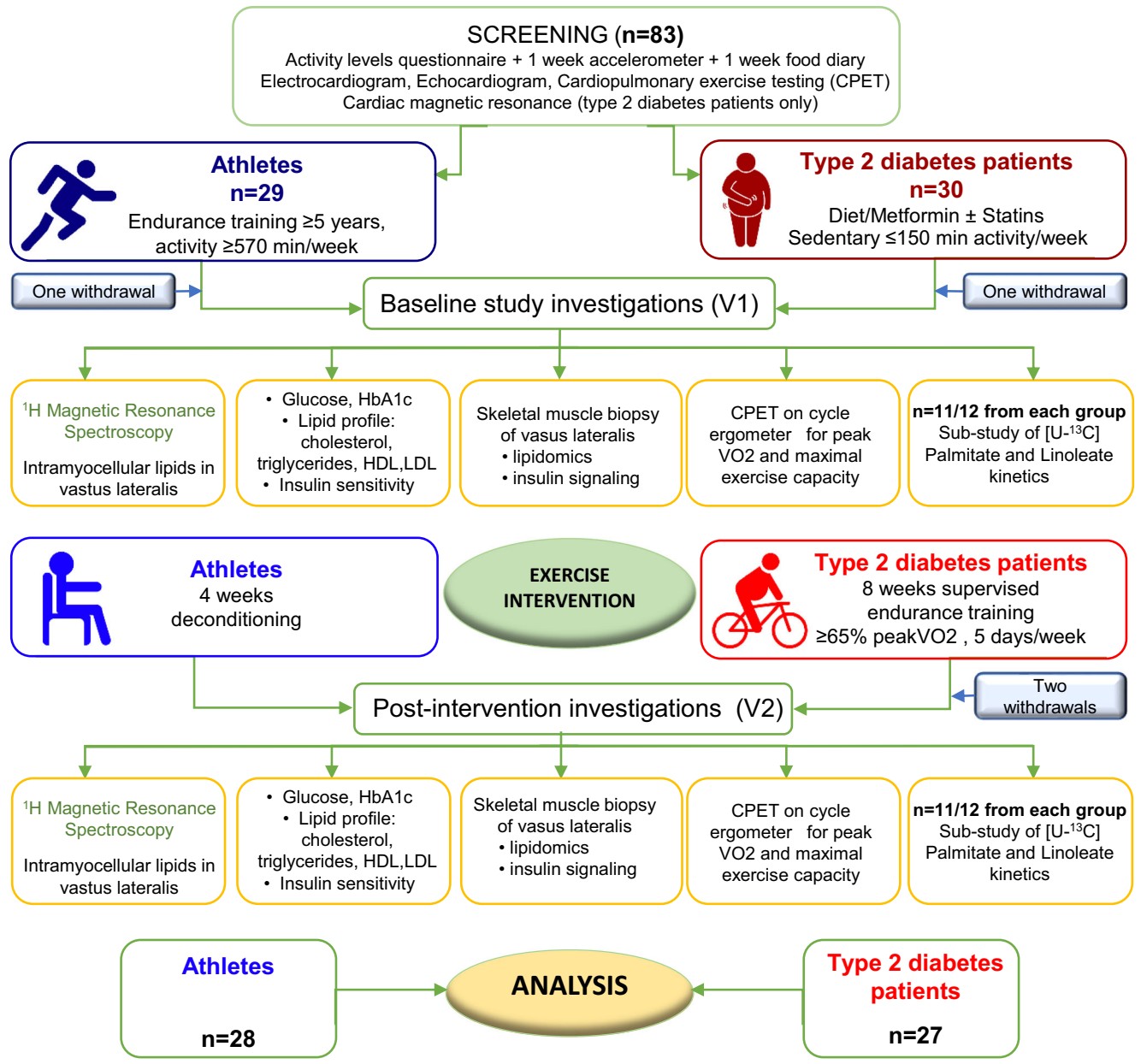

**Fig. 1 | Study consort and flow diagram, showing selection of the two study populations after screening, baseline study investigations (V1), exercise interventions and post-intervention study investigations (V2).** HbA1C— haemoglobin A1C; HDL−high-density lipoprotein; LDL−low-density lipoprotein; VO₂−oxygen consumption; CPET−cardiopulmonary exercise test.

accelerometer data whereas patients with type 2 diabetes were directly supervised by the investigators during each daily 60 min of cycling, 5 days/week for 8 weeks. To exclude any dietary change that could have been a confounder during the interventions, a repeat 1-week food diary during exercise interventions was undertaken. This showed no change in nutritional intake in either group, except for a decrease in sugar consumption by athletes (Supplemental Table 2), which paralleled the removal of structured training.

## Deconditioning in athletes

Athletes gained an average of 1.2 kg in weight following deconditioning (Table 1). However, $^1$H-magnetic resonance spectroscopy detected no significant changes in total intramyocellular lipid mass or fractions of saturated/unsaturated intramyocellular lipids (post deconditioning $^1$H-magnetic resonance spectroscopy intramyocellular lipid mass of

$0.038 \pm 0.01$ arbitrary units, fraction of saturated and unsaturated storage of intramyocellular lipids of $86.69 \pm 4\%$ and $13.30 \pm 4\%$, respectively, Fig. 2B). Post-deconditioning, there was an apparent trend towards a decrease in palmitate fractional incorporation rate (to $0.35 \pm 0.3\%/h$), but not for linoleate (which was measured at $0.21 \pm 0.2\%/h$) (Table 3, Fig. 3A) The rates of plasma appearance for both palmitate and linoleate were unchanged and therefore unaffected by the post-exercise intervention status. Skeletal muscle biopsies identified no changes in total, saturated or unsaturated triacylglycerols, diacylglycerols and ceramides or basal state assessment of insulin receptor pathways (Table 4). Athletes' circulating LDL-cholesterol increased with deconditioning, but their basal insulin sensitivity was maintained (Table 5). As expected, their exercise performance declined, registering less metabolic equivalents and lower peak oxygen consumption, Table 1.

**Table 1 | Weight and cardio-pulmonary exercise test performance in athletes and patients with type 2 diabetes mellitus before and after exercise interventions**

| | Baseline difference between athletes and type 2 diabetes patients | p-value | Athletes change between baseline and deconditioning | p-value | Patients with type 2 diabetes change between baseline and exercise training | p-value | Difference in changes between athletes and type 2 diabetes patients | p-value |
|---|---|---|---|---|---|---|---|---|
| Weight, kg | −20 (−27 to −14) | <0.001 | 1.2 (0.3–2.1) | 0.01 | −2.6 (−3.6 to −1.6) | <0.001 | 3.8 (2.4–5.0) | <0.001 |
| RER | 0.005 (−0.05 to 0.06) | 0.9 | 0.03 (−0.01 to 0.06) | 0.2 | −0.05 (−0.08 to −0.01) | 0.01 | 0.07 (0.02–0.1) | 0.005 |
| VO$_2$ peak, mL/min/kg | 22 (19–24) | <0.001 | −3.5 (−4.8 to −2.3) | <0.001 | 6.4 (5.4–7.4) | <0.001 | −10 (−12 to −8) | <0.001 |
| VO$_2$ at AT, mL/min/kg | 16 (14–18) | <0.001 | −3.8 (−5.3 to −2.3) | <0.001 | 5.8 (4.6–7.0) | <0.001 | −10 (−12 to −8) | <0.001 |
| VO$_2$/HR, mL/beat | 6.2 (4.5–7.9) | <0.001 | −2.1 (−2.7 to −1.4) | <0.001 | 2.7 (2.3–3.2) | <0.001 | −5 (−5.5 to −4.0) | <0.001 |
| METS | 6.2 (5.6–6.9) | <0.001 | −1.1 (−1.4 to −0.8) | <0.001 | 1.75 (1.4–2.1) | <0.001 | −2.8 (−3.2 to −2.4) | <0.001 |
| Resting Heart Rate, bpm | −7 (−13 to 1.2) | 0.02 | 1.5 (−2.3 to 5.3) | 0.4 | 2.1 (−1.2 to 5.4) | 0.2 | −0.62 (−5.6 to 4.3) | 0.8 |
| Peak Heart Rate, bpm | 11 (4–19) | 0.003 | 4.6 (1.6 to 7.7) | 0.004 | 7.4 (2.5–12.3) | 0.005 | −2.8 (−8.3 to 2.8) | 0.3 |
| Resting systolic BP, mmHg | −5 (−10 to 1.0) | 0.10 | −3.2 (−6.9 to 0.6) | 0.09 | −2.0 (−6.1 to 2.3) | 0.4 | −1 (−6.7 to 4.2) | 0.7 |
| Resting diastolic BP, mmHg | −7.3 (−12 to −3) | 0.003 | 1.9 (−1.2 to 5.0) | 0.2 | −2.8 (−5.4 to −0.17) | 0.04 | 4.7 (1–9) | 0.02 |
| Peak systolic BP, mmHg | −2 (−10 to 7) | 0.7 | −8.5 (−16 to −1) | 0.03 | 12 (6.0–17) | <0.001 | −20 (−29 to −11) | <0.001 |
| Peak diastolic BP, mmHg | −4.0 (−11 to 3) | 0.3 | −4.6 (−8.6 to −0.7) | 0.02 | −1.4 (−7 to 4) | 0.6 | −3 (−10 to 4) | 0.34 |
| Heart Rate recovery, bpm | 3.9 (−1 to 8.8) | 0.1 | −1.6 (−3.6 to 0.5) | 0.1 | 3.6 (0.5–6.6) | 0.02 | −5 (−8.7 to −1.5) | 0.006 |
| VE/VCO$_2$ slope | −2.2 (−4.1 to −0.4) | 0.019 | −0.5 (−1.6 to 0.6) | 0.4 | −0.8 (−2.0 to 0.3) | 0.1 | 0.34 (−1.2 to 1.8) | 0.7 |
| Resting O$_2$ saturation, % | 0.3 (−0.1 to 0.7) | 0.1 | −0.1 (−0.4 to 0.2) | 0.5 | 0.2 (−0.25 to 0.7) | 0.4 | −0.30 (−0.81 to 0.204) | 0.24 |
| Peak O$_2$ saturation, % | −0.1 (−1.4 to 1.3) | 0.9 | −0.5 (−1.6 to 0.6) | 0.4 | 0.3 (−1.2 to 1.7) | 0.7 | −0.8 (−2.5 to 1.0) | 0.4 |
| FEV1, L | 0.7 (0.3–1) | <0.001 | | | | | | |
| Max power output, Watt | 118 (96–140) | <0.001 | −16 (−20 to −11) | <0.001 | 35 (30–41) | <0.001 | −51 (−58 to −44) | <0.001 |
| Exercise duration, seconds | 86 (41–131) | <0.001 | −8.6 (−26 to 9) | 0.3 | 88 (72–104) | <0.001 | −97 (−120 to −73) | <0.001 |

Data are shown as mean and 95% confidence intervals of inter-or intra-group differences/changes parametric hypothesis tests.

Athletes n = 28; type 2 diabetes patients n = 27.

RER respiratory exchange ratio, VO$_2$ oxygen consumption, AT anaerobic threshold, VO$_2$/HR oxygen uptake per heartbeat or 'oxygen pulse', MET metabolic equivalents, HR heart rate, BP blood pressure, VE/VCO$_2$ ventilatory efficiency, FEV1 forced expiratory volume in first second.

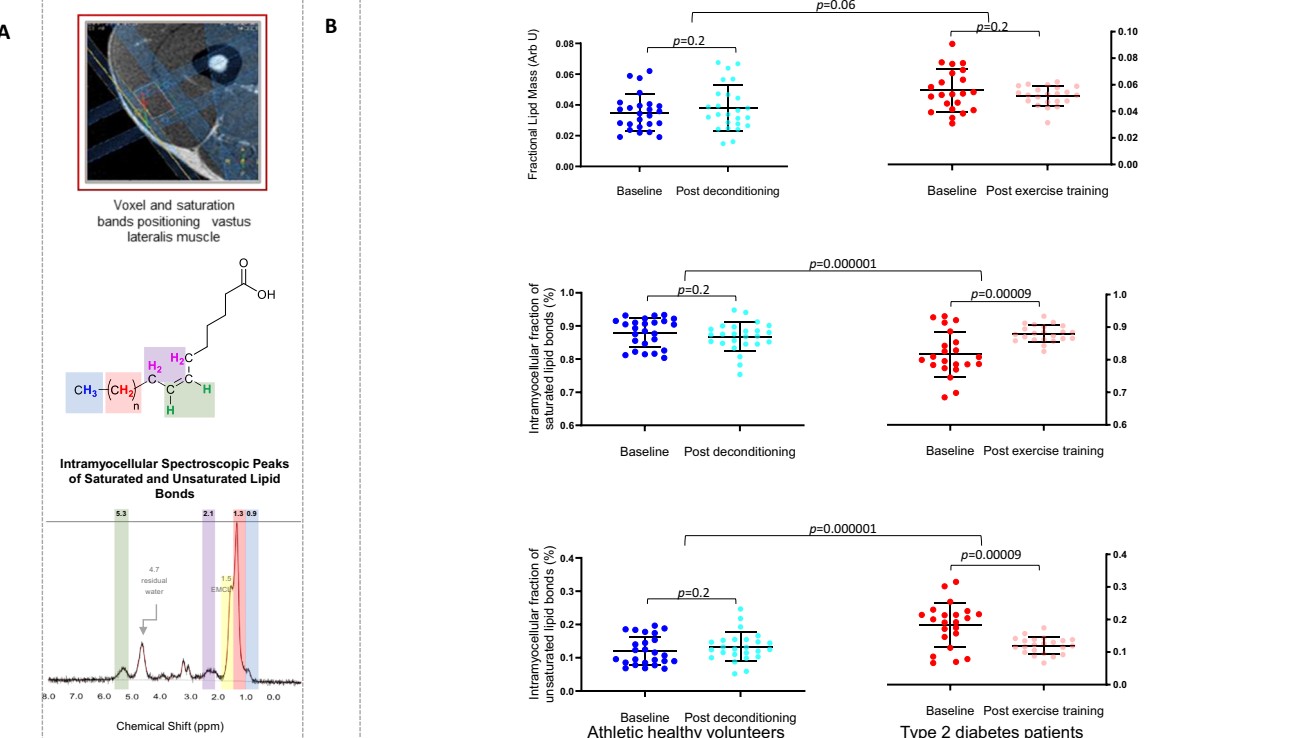

**Fig. 2 | ¹H-Magnetic resonance spectroscopy of total, saturated and unsaturated intramyocellular lipid bonds. 2A**: Top: Representative localisation of the spectroscopy voxel for ¹H-magnetic resonance of the right vastus lateralis. Middle: Colour-coded schematic showing protons connected to carbon nuclei in single (saturated) covalent bonds in blue and red and protons connected to double/triple (unsaturated) carbon bonds or adjacent to double/triple bonds, which are shown in green and purple, respectively. Bottom: The same colour-coding shows how the different proton species process at different frequencies in the magnetic field, generating different spectral peaks as shown by the different chemical shifts (parts per million, ppm) of saturated (0.9 ppm, 1.3 ppm) and unsaturated (2.1 ppm, 5.2 to 5.3 ppm) intramyocellular lipid peaks. Main extramyocellular lipid (EMCL) peak is separated at 1.5 ppm (yellow). **2B**: Intramyocellular total lipids and fractions of saturated and unsaturated carbon bonds within intramyocellular lipid (n = 25 athletes and n = 22 patients with type 2 diabetes), Arb U = arbitrary units. Data is shown as individual data points with means and error bars for standard deviation. Baseline and post-interventions comparisons between groups were performed using t-tests. Top: Intra-myocellular fractional lipid mass in athletes and type 2 diabetes patients before and after exercise intervention Middle: Intra-myocellular fraction of saturated lipids in athletes and type 2 diabetes patients before and after exercise intervention. Bottom: Intra-myocellular fraction of unsaturated lipids in athletes and type 2 diabetes patients before and after exercise intervention.

## Endurance exercise training in patients with type 2 diabetes

Patients lost an average of 2.6 kg in weight following exercise training, Table 1. Although there were no demonstrable changes in total intramyocellular lipid mass detected with ¹H-magnetic resonance spectroscopy (0.052 ± 0.007 arbitrary units), the fraction of intramyocellular saturated lipid increased (to 88.00 ± 3%), with a reciprocal fall in unsaturated lipids (to 12.00 ± 3%), Fig. 2B. This was accompanied by an increase in both palmitate and linoleate fractional incorporation rate [to 0.35 ± 0.3%/h (2.4-fold) for palmitate and to 0.21 ± 0.2%/h (1.7-fold) for linoleate], Table 3 and Fig. 3A, although the rates of plasma appearance for both palmitate and linoleate were also unchanged and therefore unaffected by the post-exercise intervention status. Skeletal muscle biopsies identified no changes in total, saturated or unsaturated triacylglycerols, diacylglycerols and ceramides (Table 4) but there was an increase in the ratios of basal state phosphorylated/total protein kinase B (p/t AKT), and phosphorylated/total protein levels of 5' adenosine monophosphate-activated protein kinase (p/t AMPK) (Table 4 and Fig. 3B) consistent with enhanced post insulin receptor pathway and metabolic sensing. There were additional health improvements in achieving lower serum cholesterol, triglycerides, fasting glucose concentrations as well as HBA1c, which was associated with an improvement of their basal insulin sensitivity (Table 5). Remarkably, after exercise training patients with type 2 diabetes achieved lower serum cholesterol and LDL-cholesterol compared to athletes, although their fasting glucose and plasma non-esterified (free) fatty acids remained higher and serum HDL-cholesterol lower.

Their exercise performance improved, achieving longer exercise duration, more metabolic equivalents and higher peak oxygen consumption compared with before training (Table 1).

Extramyocellular skeletal muscle lipids and subcutaneous adipose tissue were also investigated with ¹H-magnetic resonance spectroscopy with no within or between group differences before or after either exercise intervention (Supplemental Fig. 1, 2).

## Discussion

This parallel mechanistic skeletal muscle phenotyping study identified structural and functional differences in intramyocellular lipids between athletes and type 2 diabetes patients. Compared to patients with type 2 diabetes, athletes have a higher saturation of intramyocellular lipid storage as well as a 4.1-fold and 2.3-fold higher turnover of saturated and unsaturated fatty acids respectively. Deconditioning of athletes resulted in no alterations of their skeletal muscle phenotype except a trend in lowering their very high saturated (palmitate) lipid turnover which was a characteristic of their peak exercise performance. Conversely, 8-week endurance exercise training for patients with type 2 diabetes resulted in numerous important re-adaptations. There was an increase in intramyocellular lipid saturation and a larger increase in saturated compared to unsaturated skeletal muscle lipid turnover. These intramyocellular modifications aligned the diabetes skeletal muscle phenotype to that of deconditioned athletes. There were concomitant improvements in basal insulin sensitivity, serum cholesterol and triglycerides, glycaemic control, and physical

**Table 2 | Total, saturated and unsaturated intramyocellular lipid bonds in athletes and Type 2 diabetes patients before and after exercise interventions**

| | Baseline differences between athletes and type 2 diabetes patients | p-value | Athletes change between baseline and deconditioning | p-value | Patients with type 2 diabetes change between baseline and exercise training | p-value | Difference in changes between athletes and type 2 diabetes patients | p-value |
|---|---|---|---|---|---|---|---|---|
| Intramyocellular fractional lipid mass (Arb U) | −0.02 (−0.028 to −0.0125) | <0.001 | 0.003 (−0.001 to 0.007) | 0.2 | −0.004 (−0.01 to 0.002) | 0.2 | 0.007 (−0.0003 to 0.01) | 0.06 |
| Fraction of saturated carbon bonds in intramyocellular lipids (%) | 0.06 (0.03–0.09) | <0.001 | −0.01 (−0.03 to 0.005) | 0.2 | 0.06 (0.04 to 0.08) | <0.001 | −0.076 (−0.1 to −0.04) | <0.001 |
| Fraction of unsaturated carbon bonds in intramyocellular lipids (%) | −0.06 (−0.09 to −0.03) | <0.001 | 0.01 (−0.005 to 0.03) | 0.2 | −0.06 (−0.08 to −0.04) | <0.001 | 0.076 (0.48 to 0.1) | <0.001 |

Data are shown as mean and 95% confidence intervals of inter-or intra-group differences/changes parametric hypothesis tests.
Athletes *n* = 25; type 2 diabetes patients *n* = 25.
*Arb U* arbitrary units.

**Table 3 | Fractional incorporation rate (FIR) and rate of appearance (Ra) of U$^{13}$C- potassium palmitate (16.0) and U$^{13}$C-potassium linoleate (18.2 n-6) of athletes and patients with type 2 diabetes mellitus before and after exercise interventions**

| | Baseline differences between athletes and type 2 diabetes patients | p-value | Athletes change between baseline and deconditioning | p-value | Patients with type 2 diabetes change between baseline and exercise training | p-value | Difference in changes between athletes and type 2 diabetes patients | p-value |
|---|---|---|---|---|---|---|---|---|
| FIR 16.0 (%/h) | 0.46 (0.18–0.76) | 0.003 | −0.27 (−0.57 to 0.3) | 0.07 | 0.2 (0.002–0.4) | 0.04 | −0.5 (−0.8 to −0.1) | 0.008 |
| FIR 18.0 n-6 (%/h) | 0.15 (0.05–0.24) | 0.003 | −0.06 (−0.16 to 0.05) | 0.3 | 0.095 (0.1–0.17) | 0.02 | −0.15 (−0.27 to −0.02) | 0.01 |
| Ra 16.0 (mmol/h) | 0.7 (−0.9 to 2.3) | 0.4 | −0.8 (−2.2 to 0.7) | 0.2 | −0.09 (−1.3 to 1.1) | 0.9 | −0.7 (−2.4 to 1.0) | 0.4 |
| Ra 18.2 n-6 (mmol/h) | −0.6 (−2.2 to 1.0) | 0.4 | −0.234 (−1.1 to 0.6) | 0.5 | −0.204 (−1.2 to 0.8) | 0.7 | −0.03 (−1.3 to 1.2) | 0.9 |

Data are shown as mean and 95% confidence intervals of inter-or intra-group differences/changes parametric hypothesis tests.
Athletes *n* = 11; type 2 diabetes patients *n* = 12.
*FIR* fractional incorporation rate, *C16.0* U$_{13}$C- potassium palmitate, *C18.2 n-6* U$^{13}$C-potassium linoleate, *Ra* rate of appearance, *h* hour.

performance. After endurance exercise training, patients with type 2 diabetes also demonstrated upregulation in basal state AKT (S473) and AMPKα (T172) phosphorylation, suggesting that both insulin receptor and metabolic pathways were sensitised basally within skeletal muscle cells.

Our findings show that maladaptive intramyocellular skeletal muscle changes seen in the early stages of type 2 diabetes can be successfully reversed to a great extent. These data support the concept that skeletal muscle insulin resistance could be the primary defect in type 2 diabetes, as some have proposed[15,16]. Although further work would be required to elucidate the causality between muscle insulin resistance and intramyocellular lipid accumulation, these findings suggest that saturation of the ectopic intramyocellular lipid accumulation could be a new target for health improvement in type 2 diabetes. The concept that saturated fat can be beneficial, especially when stored, challenges the traditional view that associates all saturated fat with an increased risk of cardiovascular disease and worse clinical outcomes.

Structured physical exercise is a cheap but effective lifestyle intervention with powerful prognostic outcomes proven in general populations, and patients with cardiac conditions or diabetes[17,18]. Due to known sex differences in intramyocellular lipid stores and utilisation rates (both higher in women[19]) in this initial exploration we studied only male participants to maintain a reasonable sample size for such an intensive study protocol. To age-match the groups we recruited recreational competitive athletes according to the European Society of Cardiology definition[20], involved in endurance training, as elite athletes do not exist in this age group.

Our clinical trial reveals that, contrary to its known bad press, *saturated* fat is in fact essential for high level performance of skeletal muscle and its reduced intramyocellular availability or utilisation is tracking with the insulin resistant status and metabolic dysfunction of people with type 2 diabetes. The skeletal muscle lipidomic analysis detected no baseline or exercise-induced differences in total levels of saturated versus unsaturated tri/di-acylglycerols or ceramides between athletes and patients with type 2 diabetes. This apparent difference between the spectroscopic and lipidomic evaluations is rooted in two important technical aspects: firstly, spectroscopy assesses proportions of saturated/unsaturated carbon bonds present only in intramyocellular lipid stores, whereas lipidomics reports whole fatty acids per se, which are present in the intra- as well as extramyocellular compartments, including the cell membranes. Saturated fatty acids generally represent a much smaller component of the skeletal muscle lipid composition compared to the unsaturated ones, as for any given carbon length there is only one saturated fatty acid and numerous unsaturated ones. However, our MRI assessment of saturated/unsaturated carbon bonds shows that the saturated fraction of the intramyocellular lipid storage is much higher in athletes compared to patients with type 2 diabetes.

Given the dogma that saturated fat is harmful mainly through its contribution to atherosclerosis-related pathology, it is rather surprising that intramyocellular lipid saturation is reduced in patients with type 2 diabetes. Although the total intramyocellular pool did not reduce in size after patients' endurance training, its composition did alter, with increased saturation after exercise training, at least for the

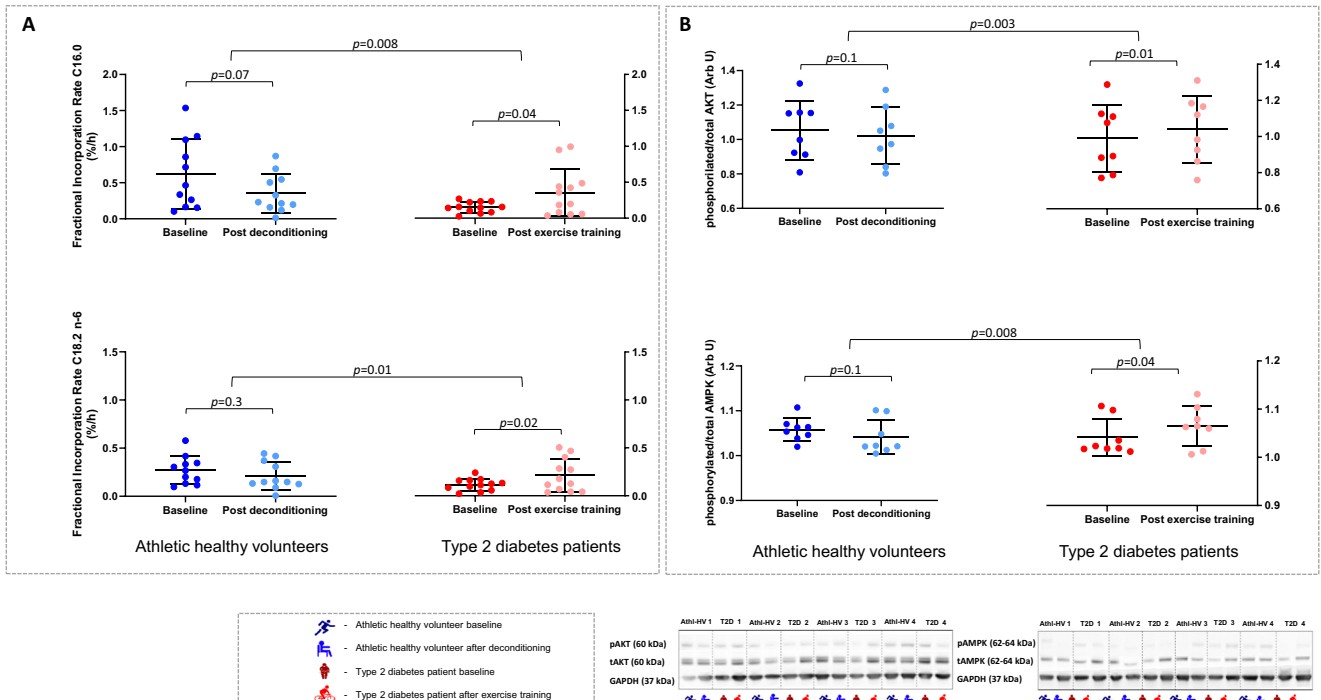

**Fig. 3 | Skeletal muscle saturated and unsaturated lipid turnover and insulin receptor pathway/metabolic sensing.** Data are shown as individual data points with means and error bars for standard deviation. Arb U arbitrary units. **3A:** Top: Fractional incorporation rate for [U-$^{13}$C] Palmitate (16:0) before and after exercise intervention in athletes and patients with type 2 diabetes ($n = 11$ athletes and $n = 12$ patients with type 2 diabetes) Baseline and post-interventions comparisons between groups were performed using $t$ tests. Bottom: Fractional incorporation rate for [U-$^{13}$C] Linoleate (18.2 n-6) before and after exercise intervention in athletes and patients with type 2 diabetes ($n = 11$ athletes and $n = 12$ patients with type 2

diabetes). **3B:** Top: Ratio of phosphorylated protein kinase B (AKT) to total of AKT (p/t AKT) protein levels before and after exercise intervention in athletes and patients with type 2 diabetes ($n = 8$ in each group). Baseline and post-interventions comparisons between groups were performed using $t$ tests. Bottom: Ratio of phosphorylated 5' adenosine monophosphate-activated protein kinase (AMPK) to total of AMPK (p/t AMPK) protein levels before and after exercise intervention in athletes and patients with type 2 diabetes ($n = 8$ in each group). Below are representative western blot examples from $n = 4$ in each group.

duration of exercise taken during our study. One possible explanation for these findings is that compared to saturated fatty acid metabolism, β-oxidation of unsaturated fatty acids requires two additional energy-consuming catalytic reactions, one by an isomerase and one by a reductase enzyme[21,22]. Therefore, overall, β-oxidation of an unsaturated fatty acid with the same chain length as a saturated fatty acid will yield less energy[23]. Whilst it is understandable that with training, the skeletal muscle of healthy athletes will tend to store its fuel in a more efficient way (i.e. saturated fat), as it yields more energy, it is less clear why insulin resistant status would be associated with unsaturated intramyocellular storage. This finding cannot be attributed to differences in diets or changes in nutritional intake since participants were instructed to avoid dietary changes. Even though the 1-week food diaries almost certainly under-reported their true intake, which is a well documented phenomenon[24], importantly, there was no significant change between or within groups (except the athletes' sugar drinks requirements which were discontinued during deconditioning). Therefore, we conclude that skeletal muscle lipid saturation could represent a new biomarker of metabolic health in type 2 diabetes.

Previous investigations have shown that specific lipid species (tri-, diacyl-glycerols, ceramides, sphingolipids) can influence insulin sensitivity status of healthy participants, participants with obesity and participants with type 2 diabetes[9,10]. Even though both ceramides and diacylglycerols have been identified as lipotoxic mediators of insulin resistance[25], it is increasingly more apparent that specific lipid species, or isoforms, or even the sub-cellular localisation of their saturated or unsaturated forms which confers more importance in skeletal muscle insulin sensitivity than their total quantity[10,26–28]. These species are

contained in the lipid bi-layers of sarcolemma, sarcoplasmic reticulum, mitochondrial membrane, as well as stored as ectopic intramyocellular fat and present in extramyocellular lipids which cannot be separated in a biopsy[29]. The lack of any significant change in muscle lipidomic analyses of total intramyocellular triacylglycerols, diacylglycerols or ceramides between athletes and patients with type 2 diabetes at baseline or after exercise intervention add onto a body of literature that remains at variance[30,2,31]. Notwithstanding conflicting reports of such total changes, further work will need to identify species, isoforms[26,28] or subcellular localisations[10] for example their proximity to the mitochondria, which can be modulated as key players responsible for cardio-metabolic health improvements.

We investigated the turnover of the two most abundant saturated (palmitate) and unsaturated (linoleate) fatty acids, the latter a first-in-man exploratory application of fat metabolism. As expected from previous studies[6], trained athletes had higher turnover of both palmitate and linoleate compared to patients with type 2 diabetes, being heavily reliant on palmitate utilisation for high physical performance, whereas in sedentary patients with type 2 diabetes, the palmitate turnover was drastically reduced with rates comparable to those of linoleate. Remarkably, improvements in basal insulin sensitivity through chronic aerobic exercise in patients with type 2 diabetes were also accompanied by a re-arrangement of their intramyocellular lipid turnover which re-aligned their phenotype with that of the deconditioned athletes. The structural and metabolic plasticity demonstrated in this study holds promise for further remedial interventions in diabetes but cannot of course be extrapolated to more advanced forms of disease or those on multiple hypo-glycaemic agents. It is remarkable that exercise training alone in patients with type 2 diabetes

**Table 4 | Skeletal muscle saturated/unsaturated main lipid species and basal insulin receptor/metabolic pathway of athletes and patients with type 2 diabetes mellitus before and after exercise interventions**

| | Baseline differences between athletes and type 2 diabetes patients | p-value | Athletes change between baseline and deconditioning | p-value | Patients with type 2 diabetes change between baseline and exercise training | p-value | Difference in changes between athletes and type 2 diabetes patients | p-value |
|---|---|---|---|---|---|---|---|---|
| *LIPIDOMICS (pmol/mg protein)* | | | | | | | | |
| Total TAG | −65349 (−224862 to 94164) | 0.4 | 110922 (−27589 to 249433) | 0.1 | −8724 (−234766 to 217319) | 0.9 | 119645 (−126564 to 365856) | 0.3 |
| Saturated TAG | −392 (−1621 to 837) | 0.5 | 1102 (−537 to 2741) | 0.2 | 2210 (−3177 to 7597) | 0.4 | −1108 (−6337 to 4121) | 0.7 |
| Unsaturated TAG | −94925 (−265681 to 75831) | 0.3 | 182933 (−29276 to 395143) | 0.08 | 206250 (−328292 to 740792) | 0.4 | −23317 (−557448 to 510815) | 0.9 |
| Total DAG | −698 (−2396 to 1001) | 0.4 | 104 (−935 to 1142) | 0.8 | −245 (−2391 to 1901) | 0.8 | 348 (−1865 to 2562) | 0.7 |
| Saturated DAG | −448 (−1410 to 514) | 0.3 | 30 (−436 to 496) | 0.9 | −251 (−1391 to 888) | 0.6 | 281 (−861 to 1425) | 0.6 |
| Unsaturated DAG | −250 (−1032 to 532) | 0.5 | 75 (−538 to 688) | 0.8 | 77 (−954 to 1108) | 0.9 | −1.8098 (−1115.81 to 1112.2) | 0.9 |
| Total Ceramides | −58 (−286 to 170) | 0.6 | 4 (−97 to 105) | 0.9 | −17 (−290 to 256) | 0.9 | 21 (−249 to 292) | 0.8 |
| Saturated Ceramides | −38 (−197 to 121) | 0.6 | 22 (−65 to 109) | 0.6 | 138 (−244 to 520) | 0.4 | −115 (−479 to 248) | 0.5 |
| Unsaturated Ceramides | −11 (−84 to 63) | 0.8 | 13 (−50 to 76) | 0.7 | 39 (−102 to 180) | 0.5 | −26.4 (−170 to 117) | 0.7 |
| *INSULIN RECEPTOR PATHWAY (basal state)* | | | | | | | | |
| GLUT1/GAPDH | −0.1 (−1.3 to 1.05) | 0.8 | 0.17 (−0.01 to 0.35) | 0.06 | 0.08 (−0.09 to 0.3) | 0.3 | 0.08 (−0.14 to 0.3) | 0.4 |
| GLUT4/GAPDH | −0.07 (−1.5 to 1.3) | 0.9 | 0.01 (−0.09 to 0.1) | 0.8 | −0.03 (−0.1 to 0.07) | 0.5 | 0.04 (−0.09 to 0.2) | 0.5 |
| pIR/GAPDH (Y1162) | −0.1 (−1.1 to 0.8) | 0.7 | 0.13 (−0.009 to 0.3) | 0.06 | −0.06 (−0.2 to 0.09) | 0.4 | 0.2 (0.005 - 0.4) | 0.04 |
| pIRS-1Ser612/GAPDH | −0.06 (−1.5 to 1.3) | 0.9 | −0.04 (−0.15 to 0.07) | 0.4 | −0.11 (−0.2 to 0.006) | 0.06 | 0.069 (−0.07 to 0.2) | 0.3 |
| pS6/GAPDH | −0.1 (−0.6 to 0.4) | 0.7 | 0.07 (−0.006 to 0.15) | 0.07 | −0.03 (−0.1 to 0.09) | 0.6 | 0.1 (−0.03 to 0.2) | 0.1 |
| pERK/GAPDH | −0.06 (−1.3 to 1.2) | 0.9 | −0.096 (−0.3 to 0.15) | 0.4 | −0.13 (−0.3 to 0.05) | 0.1 | 0.04 (−0.2 to 0.3) | 0.7 |
| p/t AKT | 0.06 (−0.12 to 0.25) | 0.5 | −0.03 (−0.07 to 0.010) | 0.1 | 0.05 (0.014 to 0.08) | 0.01 | −0.07 (−0.1 to −0.03) | 0.003 |
| p AKT Ser 473 | −37 (−7693 to 7618) | 0.9 | 125 (−343 to 593) | 0.6 | −120 (−532 to 292) | 0.5 | 246 (−320 to 812) | 0.4 |
| t AKT | −3750 (−8072 to 572) | 0.08 | 1849 (−611 to 4309) | 0.1 | −2811 (−5210 to −413) | 0.03 | 4661 (1544 to 7777) | 0.006 |
| p/t AMPK | 0.02 (−0.01 to 0.05) | 0.3 | −0.01 (−0.04 to 0.004) | 0.1 | 0.02 (0.0007 – 0.05) | 0.04 | −0.04 (−0.07 to −0.01) | 0.008 |
| p AMPK T172 | −286 (−967 to 395) | 0.4 | 233 (−208 to 674) | 0.3 | −40 (−164 to 83) | 0.5 | 273 (−142 to 688) | 0.1 |
| t AMPK | −1291 (−3468 to 886) | 0.2 | 1242 (−95 to 2579) | 0.06 | −1416 (−2880 to 49) | 0.06 | 2658 (859 - 4457) | 0.007 |

Data are shown as mean and 95% confidence intervals of inter-or intra-group differences/changes parametric hypothesis tests.
Lipidomics: Athletes *n* = 10; type 2 diabetes patients *n* = 11 insulin receptor pathway: Athletes *n* = 8; type 2 diabetes patients *n* = 8.
*TAG* triacylglycerol, *DAG* diacylglycerol, *GLUT1* glucose transporter 1, *GAPDH* glyceraldehyde 3-phosphate dehydrogenase, *GLUT4* glucose transporter 4, *pIR* phosphorylated insulin receptor, *pIRS-1* phosphorylated insulin receptor substrate-1, *PS6* phosphorylated ribosomal protein S6, *pERK* phosphorylated extracellular signal-regulated kinase, *p/t AKT* phosphorylated over total protein kinase B, *p AKT* phosphorylated protein kinase B, *t AKT* total protein kinase B, *p/t AMPK* phosphorylated over total AMP-activated protein kinase, *p AMPK* phosphorylated AMP-activated protein kinase, *t AMPK* total AMP-activated protein kinase.

significantly increases basal phosphorylation of AMPKα (T172) and Akt (S473) (in keeping with their improvements in systemic basal insulin sensitivity post-exercise training), even though trans-membrane glucose transporters GLUT1/4, IRS, pS6 or pERK remained unchanged after training. This re-emphasises the role of subcellular localisation and saturation of specific lipid fractions, such as diacylglycerols and ceramides[10,32]. The skeletal muscle AMPK activation (increased phosphorylation) in basal state suggests that the energy-sensing mechanism is upregulated in exercise-trained patients with type 2 diabetes, in keeping with their increased fatty acid fluxes. These findings complement previous observations that intramyocellular lipid droplets' morphology and their sub-cellular distribution in type 2 diabetes patients also showed a phenotype shift towards an athlete-like appearance after endurance training[33].

Although we demonstrated significant beneficial changes in abundance and turnover of saturated intramyocellular lipids in response to exercise training in type 2 diabetes patients, physical exercise is not a practical solution for every individual. Further work should seek to explore which exercise-mimetic pharmacotherapies are capable of recapitulating these findings in type 2 diabetes patients. In particular, solutions targeting insulin resistance through specific intramyocellular lipids or indirectly via upstream nutrient-sensing transcription factors[34] targeting genes involved in specific lipid biosynthesis (such as long-chain fatty acid elongase 6[35] or Stearoyl-CoA desaturase[36] respectively) could be explored.

Limitations: We did not use a skeletal muscle-specific insulin resistance test or a hyperinsulinaemic-euglycaemic clamp for several pragmatic reasons, of added clinical risk, study visit duration, and complexity. For the same reasons of subject safety (avoiding a too prolonged fasting period) as well as pragmatically completing study visits within 8 h, we elected to measure insulin sensitivity by QUICKI and HOMA-IR indices rather than a more accurate oral glucose tolerance test. Our lipidomic analysis was a semi-quantitative approach aimed at detecting relative differences only. Whilst no medication was discontinued in the type 2 diabetes patients for the study duration, it is important to recognise that both metformin and statins use can result in a reduction or increase, respectively, in insulin resistance status. An influence on the results from nutritional intake cannot be definitively excluded.

**Table 5 | Metabolic profile of athletes and patients with type 2 diabetes mellitus in basal conditions before and after exercise interventions**

| | Baseline differences between athletes and type 2 diabetes patients | p-value | Athletes change between baseline and deconditioning | p-value | Patients with type 2 diabetes change between baseline and exercise training | p-value | Difference in changes between athletes and type 2 diabetes patients | p-value |
|---|---|---|---|---|---|---|---|---|
| Cholesterol, mmol/L | 0.6 (−0.2 to 1.3) | 0.1 | 0.3 (−0.003 to 0.55) | 0.05 | −0.2 (−0.5 to 0.02) | 0.08 | 0.5 (0.1–0.9) | 0.008 |
| LDL-cholesterol, mmol/L | 0.3 (−0.3 to 0.9) | 0.3 | 0.2 (0.02 to 0.5) | 0.03 | −0.16 (−0.4 to 0.04) | 0.1 | 0.4 (0.1–0.7) | 0.007 |
| HDL-cholesterol, mmol/L | 0.6 (0.3–0.8) | <0.001 | −0.010 (−0.09 to 0.08) | 0.8 | 0.08 (−0.02 to 0.2) | 0.1 | −0.09 (−0.2 to 0.04) | 0.1 |
| Triglycerides, mmol/L | −0.9 (−1.3 to −0.4) | <0.001 | 0.1 (−0.08 to 0.3) | 0.3 | −0.5 (−0.8 to −0.13) | 0.009 | 0.6 (0.2–0.9) | 0.003 |
| NEFA, mmol/L | −0.098 (−0.19 to −0.003) | 0.04 | −0.05 (−0.13 to 0.04) | 0.3 | 0.03 (−0.05 to 0.1) | 0.5 | −0.0791 (−0.2 to 0.04) | 0.2 |
| Glucose, mmol/L | −2.8 (−3.9 to −1.7) | <0.001 | −0.017 (−0.2 to 0.17) | 0.9 | −0.96 (−1.8 to −0.17) | 0.02 | 0.9 (0.2–1.7) | 0.01 |
| Plasma Insulin, mU/L | −4.9 (−7.3 to −2.6) | <0.001 | 0.2 (−0.7 to 1.05) | 0.7 | −1.2 (−3.1 to 0.7) | 0.2 | 1.4 (−0.7 to 3.4) | 0.2 |
| HbA$_1$C (mmol/mol) | | | | | −11 (−18 to −4.3) | 0.002 | | |
| QUICKI | 0.035 (0.027–0.044) | <0.001 | −0.0010 (−0.006 to 0.004) | 0.7 | 0.0073 (0.002–0.01) | 0.006 | −0.008 (−0.02 to −0.002) | 0.02 |
| HOMA2-IR | −0.85 (−1.18 to −0.52) | <0.001 | 0.02 (−0.099 to 0.1) | 0.7 | −0.3 (−0.5 to 0.02) | 0.07 | 0.3 (−0.010 to 0.6) | 0.058 |

Data are shown as mean and 95% confidence intervals of inter-or intra-group differences/changes parametric hypothesis tests.

Athletes n = 28; type 2 diabetes patients n = 27.

LDL low-density lipoprotein, HDL high-density lipoprotein, NEFA non-esterified (free) fatty acids, QUICKI Quantitative Insulin-sensitivity CheCK Index, HOMA2-IR The homoeostasis model assessment 2 insulin resistance.

## Methods

### Study approval
The study was approved by the North of Scotland Ethics Research Committee. All participants provided written informed consent prior to screening procedures and participation in the study (Clinical Trial Registration: NCT03065140: Muscle Fat Compartments and Turnover as Determinant of Insulin Sensitivity (MISTY).

### Interventions
The purpose of this parallel non-randomised, non-blinded trial is to understand how the fat within the muscle can be changed to improve blood sugar control, ultimately to reduce the risk of developing heart disease, diabetes and stroke. During the study, 29 male patients with diabetes were investigated at baseline and at the end of the study after following a supervised endurance exercise training program for a period of 8 weeks. Thirty male endurance athletes were investigated during highly trained status at baseline and at the end of the study after following a period of deconditioning for 4 weeks.

The primary outcomes of this study were the assessment and comparison of non-invasive [1]H Magnetic Resonance Imaging of the vastus lateralis muscle to determine the saturated and unsaturated intramyocellular lipid storage before and after deconditioning (in athletes) or exercise training (in patients with diabetes). The secondary outcomes were the assessment and comparison of saturated and unsaturated lipid pool turnover examined with stable isotopes before and after deconditioning (in athletes) or exercise training (in patients with diabetes).

### Participants
Age-matched male endurance-trained recreational competitive athletes and people with type 2 diabetes were enroled and provided informed consent. Recruitment commenced on 7 September 2016 and concluded on 11 January 2019. For a study of this size, only males were considered for enrolment due to known significant sex differences in intramyocellular lipid storage and utilisation between men and women[19]). Athletes were recruited from cycling/running/triathlon clubs and had a ≥5-year history of active training in moderate-vigorous intensity ≥420 min/week (total physical activity level ≥570 min/week). Patients with type 2 diabetes were recruited from local primary care centres having been diagnosed by World Health Organisation criteria of fasting glucose and 2-h oral glucose tolerance test or HbA1c ≥ 48 mmol/mol, were treated by diet ± metformin and were sedentary. Exclusion criteria for patients were: known coronary artery disease, untreated endocrine conditions, estimated glomerular filtration rate <60 mL/min/1.73m², blood pressure ≥ 160/100 mmHg, medication influencing glucose or fatty acid metabolism, such as peroxisome proliferator-activated receptors α- or γ-agonists (fibrates or thiazolidinediones respectively), niacin, angiotensin-converting enzyme inhibitor therapy, omega-3 fatty acids or any clinical condition that, in the judgement of the investigators, may have interfered with exercise intervention, fatty acid metabolism or compromise the safety of the subject.

At screening, physical activity levels were assessed using the international physical activity questionnaire (IPAQ) and objectively, with a 7-day accelerometer (wGT3X-BT ActiGraph (Actigraph, USA)) to establish habitual baseline activity. All participants completed a one-week food diary and underwent ECG, echocardiography, cardiopulmonary exercise testing, and patients had gadolinium-enhanced cardiac magnetic resonance to exclude any previous myocardial infarction[14]. Participants were studied at baseline and after an exercise intervention (Fig. 1). Both study visits took place in the morning, after a 12-h fast and all study procedures were performed at the University of Aberdeen.

Each study visit comprised of [1]H-magnetic resonance spectroscopy, skeletal muscle biopsy, venous blood sampling and cardiopulmonary exercise testing. Eleven athletes and 12 patients with type 2

diabetes underwent a 4-h stable isotope infusion followed by repeat skeletal muscle biopsy. To eliminate any possibility of the isotope sub-study results being influenced by the cardiopulmonary exercise test-ing, this was performed as the final procedure at the end of each study visit in the sub-study participants.

## Non-invasive quantification of intramyocellular lipids by ${}^1$H-magnetic resonance spectroscopy (${}^1$H-MRS)

This was performed on the right vastus lateralis using a 3 T scanner with a 16-channel receive-only torso coil (Philips Achieva MX; Phillips Healthcare, Best, NL). Participants were positioned supine, with the right leg as parallel to the magnetic field as possible[37], to optimise separation of intra- and extra-myocellular lipids. A skeletal muscle voxel was positioned to avoid extramyocellular lipid in adipose tissue or fascia and acquired with ${}^1$H-MRS point-resolved spectroscopy (PRESS; Fig. 2A): short TE (TE/TR = 26/1500 ms, number of signals averaged = 128, voxel size 15x15x15 mm)[38,39] with variable pulse power and optimised relaxation delay water suppression[40]. Six saturation bands were interleaved with the water suppression pulses[41] and used for inner volume suppression to minimise the effect of chemical shift displacement[42,43]. $B_0$ shimming was performed using second-order pencil beam[44,45]. Commercially available spectral fitting software (LCModel Version 6.3, Oakville, ON, Canada) was used to analyse the data. Spectral peak areas related to different lipid components nor-malised to internal tissue water levels were used to calculate total intramyocellular lipid mass as well as the fractional lipid mass (fLM) [lipid/(lipid+water)] and fractions of saturated (fSL) (saturated/total) and unsaturated (fUL) (unsaturated/total) carbon bonds within intra-myocellular lipids. The lipid mass may be estimated from the weighted sum of the lipid signals with weighting factors 13 accounting for the mass for a =CH- group (Lip52 and Lip53), 14 for -CH$_2$- (Lip13 and Lip21) and 15 for the -CH$_3$ group (Lip09). Calculations were based on the following formula: TOTAL LIPID = 1/2 * (Lip13 + Lip21) * 14 + (Lip52+Lip53) * 13 + 1/3 x (Lip09) * 15 + 1/3 * (Lip09) * 16. The fraction of intramyocellular unsaturated lipids (fUL) was defined as the average percent of protons from double/triple bonds of all mono- and poly-unsaturated fatty acids present in the muscle. We assumed one -CH$_3$ group per side chain– therefore, the intensity of the Lip09 resonance line scales with the number of fatty acid side chains. The 2.1 ppm peak (Lip21) scales with the number of unsaturated bonds, so: fUL = (3/2 * Lip21/Lip09), where 3/2 is the correction for number of protons, (2 for -CH$_2$ group and 3 for -CH$_3$ group of Lip09). The fraction of intramyo-cellular saturated lipid (fSL) (saturated/total intramyocellular lipids) was then calculated as: fSL = 1 − fUL. fLM was expressed in arbitrary units (A.U.) and fSL and fUL as percentages.

## Skeletal muscle biopsies lipidomic, molecular biology, and iso-tope enrichment analysis

Percutaneous biopsies were obtained through an aseptic minimally invasive technique[46], using the Magnum ® Biopsy System (MG1522, Bard) with a 12-gauge core disposable biopsy needle (Magnum®, MN1210) 15 cm above the patella and from the same depth to avoid variability in muscle fibre composition[47]. The biopsy was immediately divided on ice into 3 fragments (5 mg each), snap frozen and placed in dry storage at -80 °C.

## Lipidomic analysis of skeletal muscle

Skeletal muscle lipids were extracted according to Folch et al. [48]. Briefly, tissue samples were homogenised with chloroform/methanol (2/1 Vol/Vol, ratio solvent/tissue mass 20 mL/1 g) containing 0.05% buthylydroxytolluene and while kept under nitrogen, agitated for 60 min at room temperature. The homogenate was then washed with 0.2 volume of 0.88% KCl solution. After vortexing for 10 s, the mixture was centrifuged (600 g) and the upper phase removed by siphoning. The solvent phase was filtered and evaporated under nitrogen. Lipids

extracted were then reconstituted in 300 μL methanol and kept under nitrogen atmosphere at -80C until analysis. 17:0 ceramide 12:0/12:0 DAG (Avanti Polar Lipids, Alabaster, AL, USA) and 17:0/17:0/17:0 TAG (Sigma) were added as internal standards. Lipids were analysed by liquid chromatography-mass spectrometry (LC-MS) using a Thermo Exactive Orbitrap mass spectrometer coupled to a Thermo Accela 1250 UHPLC system with a water isopropanol/acetonitrile gradient. All samples were analysed in positive ion mode over the mass to charge ratio (m/z) range 250–2000. The data sets were processed with Pro-genesis QI software (Non-linear Dynamics, Newcastle, UK). Ion signals corresponding to the accurate m/z values for individual ceramide, diacylglycerol and triacylglycerol molecular species were extracted with the mass error set to 5 ppm. Quantification was achieved by relating the raw abundance of each lipid species to the raw abundance of internal standards.

## Western Blot analysis

Skeletal muscle biopsies were homogenised in 100 μL of ice-cold radioimmunoprecipitation assay (RIPA) buffer (10 mM Tris-HCl pH 7.4, 150 mM NaCl, 5 mM EDTA pH 8.0, 1 mM NaF, 0.1% SDS, 1% Triton X-100, 1% Sodium Deoxycholate with freshly added 1 mM NaVO$_4$ and protease inhibitors) using a PowerGen 125 homo-geniser and lysates normalised to 1 μg per 1 μL. 30 μg of sample was separated by SDS-PAGE using NuPAGE 4–12% Bis–Tris midi gels (Invitrogen) in criterion cells (Bio Rad) with MOPS SDS running buffer and transferred to nitrocellulose membranes (Biorad) using criterion blotter (Biorad). Membranes were blocked and probed for proteins of interest, all primary and secondary antibodies used are shown in Table 6 and Full Une-dited Gel for Fig. 3.

## Venous blood sampling

Venous samples were collected in EDTA and serum tubes. Plasma and serum samples were isolated by centrifugation (800 g, 10 min, 4 °C) and aliquots were stored at -80 °C. Serum glucose, cholesterol, tri-glycerides, high- and low-density lipoproteins and plasma free non-esterified fatty acid concentrations were analysed on a KONELAB30 automated analyser (Thermo Fisher Scientific, Waltham, MA, USA). Serum insulin concentration was measured using ELISA Kits (Merco-dia, Uppsala, Sweden). Glycosylated haemoglobin (HbA1c) in patients with type 2 diabetes was measured in Aberdeen Royal Infirmary clinical biochemistry laboratory. Indirect indices of insulin resistance (Home-ostasis Model Assessment 2, HOMA2 -IR[49]) and sensitivity (Quantitative Insulin Sensitivity Check Index, QUICKI[50]) were calculated.

## Cardio-pulmonary exercise testing (CPET)

Maximal exercise capacity and oxygen consumption were assessed by a cycle ergometer (Ergoselect E100 K, Cosmed, Italy) and an integrated metabolic system (Quark PFT, Cosmed, Italy) on an incremental ramp protocol until volitional exhaustion.

## Fatty acid turnover−stable isotope procedures

Good manufacturing practice grade K$^+$ [U-${}^{13}$C] 16:0 and K$^+$ [U-${}^{13}$C] 18:2 n-6 salts (Cambridge Isotope Laboratories, Inc., Tewksbury, Massa-chusetts) were solubilised, tested for sterility, pyrogenicity and stabi-lity by Tayside Pharmaceuticals (Dundee, Scotland) and stored at -80 °C. On the morning of the study, each solution (49 ml for C16:0 and 39 ml C18:2 n-6) was complexed with 70 mL human albumin (Albu-norm™ 20%, Octapharma Ltd, Switzerland). Participants were admi-nistered an intravenous infusion of 245 mg K$^+$ [U-${}^{13}$C] 16:0 and 187 mg K$^+$ [U-${}^{13}$C] 18:2 n-6 over 240 min with venous sampling (10 mL) performed at 0, 30, 60, 90, 120, 150, 180, 200, 220 and 240 min in buffered sodium citrate tubes. Plasma samples were collected by centrifugation (800 g, 10 min, 4 °C), immediately aliquoted on ice in 6 tubes and stored at -80 °C.

**Table 6 | Primary and secondary antibodies used for Western blot analysis**

| Antibody | Type | Company |
|---|---|---|
| pAKT Ser 473 | primary | Cell Signalling Technology (London, UK) |
| AKT (pan) C67E7 | primary | Cell Signalling Technology (London, UK) |
| pAMPK T172 | primary | Cell Signalling Technology (London, UK) |
| AMPKα (D63G4) | primary | Cell Signalling Technology (London, UK) |
| pIR Tyr 1162 | primary | Thermo Fisher Scientific (Waltham, USA) |
| pIRS-1 (Ser 612) | primary | Cell Signalling Technology (London, UK) |
| pERK | primary | Cell Signalling Technology (London, UK) |
| PS6 | primary | Cell Signalling Technology (London, UK) |
| GLUT1 | primary | Proteintech Europe (Manchester, UK) |
| GLUT4 | primary | Abcam (Cambridge, UK) |
| GAPDH | primary | Proteintech Europe (Manchester, UK) |
| Goat anti-rabbit IgG HRP conjugate antibody | secondary for pAKT Ser 473, PS6, pAMPK T172, pERK, pIRS-1 (Ser612), pIR Tyr 1162, GLUT1, GLUT4, AKT (pan) C67E7, AMPKα (D63G4) | Enzo Biochem Inc (New York, USA) |
| Anti-mouse IgG, HRP-linked antibody | secondary for GAPDH | Cell Signalling Technology (London, UK) |

*pAKT* phosphorylated protein kinase B, *AKT (pan)* protein kinase B, *pAMPK* phosphorylated AMP-activated protein kinase, *AMPKα* AMP-activated protein kinase alpha, *pIR Tyr 1162* phosphorylated insulin receptor tyrosine 1162, *pIRS-1 (Ser612)* phosphorylated insulin receptor substrate-1 serine 612, *pERK* phosphorylated extracellular signal-regulated kinase, *PS6* phosphorylated ribosomal protein S6, *GLUT1* glucose transporter 1, *GLUT4* glucose transporter 4, *GAPDH* glyceraldehyde 3-phosphate dehydrogenase.

### Fatty acid turnover−mass spectrometry procedures

Plasma lipids were extracted from 400 μL aliquots[51] in the presence of 0.05% butylated hydroxytoluene. Non-esterified fatty acids were isolated by thin-layer chromatography on silica gel 60 plates (Merck Millipore) using elution system isohexane-diethyl ether-acetic acid (90:30:1 by vol) and FAME were prepared by acidic transesterification. Samples were reconstituted in 125 μL hexane; m/z ions of 270 and 286 were monitored for palmitic acid, and 294 and 312 for linoleic acid using electron impact gas chromatography mass spectrometry[52] with a 1-μL injection volume. Skeletal muscle biopsies (5 mg) were homogenised manually, and total lipids were extracted according to Folch et al. [48] in the presence of 0.05% butylated hydroxytoluene. FAME were prepared by transesterification with acidified methanol (1% $H_2SO_4$ in methanol) adapted from Burge et al.[53]. Briefly, extracted lipids were mixed with toluene (1.0 ml) and incubated under nitrogen for 45 min at 80 °C with 1% in methanol (2 mL). After cooling, FAME were extracted by adding hexane (2 mL) and saturated NaCl solution (2 mL). After vortexing for 1 min, organic and aqueous phases were separated by centrifugation at 1125 g for 10 min at 8 °C and the hexane layer collected and stored under nitrogen at −80 °C before analysis by gas chromatography combustion isotope ratio mass spectrometry (GCCIRMS) as $CO_2$ with ions of 44, 45, and 46 monitored (isoprime recision fitted with an isoprime GC5 interface and an Agilent 7890B GC; Elementar, Stockport, UK). Both plasma and muscle enrichment data were expressed as tracer:tracee ratios (TTR).

### Fatty acid turnover - Palmitic and linoleic fatty acid fractional incorporation rates

Dynamic isotope dilution method was used to study plasma palmitate and linoleate turnover[54]. The GCMS m + 16 and m + 18 isotopomers were used to determine the rate of appearance (Ra) of palmitate and linoleate in plasma as described previously[55]: Ra (mmol/h) = (infusion rate mmol/h * $f_1$)/(mean plasma TTR), where $f_1$ is the isotopic purity of

either m + 16 (0.816) in the palmitate infusate or m + 18 (0.779) in the linoleate infusate. The mean plasma TTR was based on samples taken from 2-4 h of infusion. The fractional incorporation rate (FIR) for skeletal muscle palmitate and linoleate was calculated using the following formula[56]: FIR (%/h) = (total $^{13}C$ at $t_4$ − total $^{13}C$ at $t_0$) * 1/(AUC plasma TTR * $f_2$)*100, where $t_4$ = time of post-infusion biopsy (4 h) and $t_0$ is time of pre-infusion biopsy (background enrichment). The area under the curve (AUC) for plasma enrichments of each fatty acid was calculated by trapezoid analysis based on the ten time points for the m + 16 and m + 18 isotopomer enrichments, respectively. The factor $f_2$ was to adjust the m + 16 (* 1.202) and m + 18 (* 1.253) values to total fatty acid $^{13}C$ supply which was calculated using selective ion monitoring of the methyl esters with summation of the m/z ion currents.

### Exercise interventions and compliance monitoring

Athletes were asked to stop all structured exercise training (limit any structured physical activity ≤ 10 min/day for four consecutive weeks). Compliance with deconditioning was monitored with a 7-day accelerometer. Patients with type 2 diabetes were enroled in an 8-week supervised endurance exercise training programme of 5 sessions/week and each session consisted of 60 min cycling (Ergoselect E100 K,Cosmed, Italy) at 65%-85% of peak oxygen consumption. Cardiopulmonary exercise testing was used fortnightly during the training period to re-test peak oxygen consumption and increase intensity of the exercise accordingly. Throughout the study period, participants were asked to not modify their nutritional habits and completed a one-week food diary during the exercise intervention.

### Statistical analysis

Statistical analyses were conducted using SPSS v29. Due to the lack of any spectroscopic data on the fractions of saturated carbon bonds in patients with type 2 diabetes, the sample size was estimated using our own data available from non-trained, young, healthy controls whose saturated intramyocellular lipids were measured at 89.3 ± 4.3%. We

aimed to detect a 10% difference between type 2 diabetes and athletes as well as a 10% change after exercise training in type 2 diabetes patients, with a standard deviation in athletes of 5; for type 2 diabetes patients we had no prior data so predicted a higher standard deviation of 14 (different duration of disease, sedentarism, higher BMI, medications). For a study power of 0.8 and significance level of 0.05, 25 participants per group were required for detecting between-group differences and 22 participants were required in the type 2 diabetes for within group changes. For the stable isotope turnover analyses, we used the palmitate fractional synthesis rate reported by Perreault et al. [57] of $0.23 \pm 0.04$/h in pre-diabetes patients. We prespecified a change of 20% as the expected effect of exercise training in type 2 diabetes patients, therefore a population mean difference =0.04, with a SD for the difference 0.04 (for studying within group changes in male participants only), study power 0.8, significance 0.05. This required a sample size of n = 10 in the diabetes group, and thus we matched the athletes' group, without prespecifying any athlete-diabetes differences. We anticipated a higher drop-out rate than the one reported[58].

Both the primary end-point (intramyocellular saturation abundance) and secondary end-points (palmitate and linoleate turnover) analysis were performed on a per protocol set of patients with valid measurements available at baseline and post interventions. Response (ie, outcome) to exercise interventions is presented as the change (variable, change) from baseline to post-intervention calculated for each participant by subtracting the baseline variable measurement from the post-intervention measurement. Two-tailed-tests assessed the significance of between athletes and type 2 diabetes patients differences in their pre-to-post intervention changes. Baseline and post-interventions comparisons between groups were performed using appropriate two tailed $t$ tests, after testing for normality. Tabulated data is presented as mean (95% confidence intervals) of within or between group changes/differences. All other variables reported were analysed in a similar manner. Any other numeric data are presented as mean and standard deviation.

Multiple comparisons were not adjusted for, as many of the differences we report in this exploratory study have clear significance that would remain after multiple adjustment, but should be considered where there is nominal statistical significance.

### Reporting summary

Further information on research design is available in the Nature Portfolio Reporting Summary linked to this article.

## Data availability

The source data and lipidomic source data generated in this study have been deposited in the Figshare Database (https://doi.org/10.6084/m9.figshare.24219934). Participant level data for the study is also available in the Figshare Database. The Lipidomics Minimal Reporting Checklist is included in the Supplemental Material. The trial protocol can be made available upon request by contacting the Corresponding Author. Requests for data should be made and these requests will be fulfilled by the corresponding author (dana.dawson@abdn.ac.uk), providing the data will be used within the scope of the originally provided informed consent. The corresponding author aims to respond to data requests within three months. The data will be made available for a year following publication of the manuscript.

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

## Acknowledgements

The authors are grateful to Mr Simon Bath, Mr Baxter Millar and Mrs Christine Alexander, clinical trial pharmacists at Tayside Pharmaceuticals, Dundee, for the GMP-clinical grade preparation of the saturated and unsaturated fatty acids stable isotopes. Thanks go to Ms Amanda Cardy, Primary Care Research Network at the University of Aberdeen for the initial screening of primary care practices for identification of potential eligible candidates for the diabetes patient group. We thank Dr Nicola Jackson for assistance with the sample processing and measurements of isotopic enrichments at the Stable Isotope Mass Spectrometry Unit, University of Surrey. We offer our thanks and gratitude to Professor Bryan Bergman and Dr Leigh Perreault from the University of Colorado, USA, for their help and advice during the isotope sub-study. The MISTY study was funded by the British Heart Foundation Project

Grant no. PG/15/88/31780, Muscle fat compartments and turnover as a determinant of insulin sensitivity, chief investigator D Dawson.

## Author contributions

D.D., A.H., S.G., S.P. and M.D. conceived the study and together with L.v.L., G.L., F.T. obtained the grant funding. AM executed the patient screening, recruitment, intervention planning, carried out all study investigations under respective specialist supervision (A.H./D.C./D.D. for magnetic resonance spectroscopy, F.T./G.L./D.D. for stable isotope investigation, S.G. for exercise intervention, S.P. for clinical supervision/management of diabetes as required, M.D. for all molecular laboratory analyses, A.M. analysed all data and performed statistical analysis under the supervision of G.H. L.v.L. provided expert advice in athletic physiology. Lipidomic analyses were carried out in P.W. laboratory. Blood/skeletal muscle enrichment analyses were carried out in B.F./F.T.-G.L. laboratories respectively, with practical input from R.G. A.R. and L.C. contributed as overall help to deliver study assessments in a technical role. M.K.H. analysed the food diaries. D.E.N. contributed to manuscript writing. D.D. and M.D. were the PhD supervisors for A.M. whose PhD thesis was based on this work. All authors contributed their respective specialist sections in drafting the manuscript.

## Competing interests

The authors declare no competing interests.
