## [Peer Review File · Nature Communications]

Comparison of intramyocellular lipid metabolism in patients with diabetes and male athletesREVIEWER COMMENTS

Reviewer #1 (Remarks to the Author):

The authors performed a study to evaluate differences in muscle lipid metabolism and storage to help explain the "athlete's paradox" in male athletes and individuals with type 2 diabetes. The focus of the study is alterations in muscle triglyceride storage, triglyceride saturation, muscle lipid accumulation, plasma and muscle FFA turnover, and insulin sensitivity. The studied athletes before and after 4 weeks of detraining, and individuals with type 2 diabetes before and after 8 weeks of endurance training. The authors found that athletes have greater storage of saturated triacylglycerol by MRI but not muscle biopsy analysis with greater palmitate kinetics that is dampened by detraining. Individuals with type 2 diabetes have more muscle triglyceride that is more unsaturated and blunted FFA kinetics, both of which are improved after training.

Major: The authors appear to have measured insulin signaling using non-insulin stimulated muscle biopsies. Therefore, these insulin signaling data are not appropriate as no differences in insulin signaling would be expected since they are only measuring basal signaling. Additionally, the specific phosphorylation sites listed in table 4 should be stated in the table, rather than forcing the reader to read supplemental methods because these phosphorylation sites are key to understanding the data shown in Table 4. For instance, for IRS1 – did the authors measure serine phosphorylation (inhibitory) or tyrosine phosphorylation (stimulatory)? This should be made more obvious to the reader. Further, the authors are trying to make conclusions on insulin signaling based on basal non-insulin stimulated muscle biopsies – which is not how these measurements are performed. To make conclusions about insulin signaling the authors would need to compare the increase in insulin signaling measurements (for instance the AKTser473 phosphorylation/total) in basal compared to insulin stimulated conditions before and after the intervention. Just comparing basal measurements of insulin stimulated insulin signaling is not appropriate for these conclusions.

The lipidomic data is concerning because it does not appear that the methods used had representative saturated and unsaturated ceramide, diacylglycerol, and triglyceride standards used to quantify saturated and unsaturated lipid species. They only appear to have saturated internal standards. Therefore, the authors are assuming that ionization and calculated abundances of both saturated and unsaturated species are identical – which is not always true and could be driving differences in the data. Ideally, the lipidomic analysis would have representative standards against which the ratio of analyte to internal standard could be quantified that would be specific for each species. Interpretation is difficult as it does not appear that the lipidomic method has been published.

There is internal inconsistency in this manuscript that needs to be addressed. There was a significant change in saturated and unsaturated TAG storage by MRI, but not by biopsy, after the exercise training intervention with individuals with type 2 diabetes. Therefore it is difficult to understand which data should be believed. The current discussion of this topic should be expanded. Further, the muscle biopsy TAG data should be calculated as percent saturation/unsaturation so that these data can be compared to the MRI data. Currently it is difficult to compare these two measurements. For instance, lines 258-259 appears to be describing differences in the TAG data by MRI, as there did not appear to be differences in saturated TAG storage from the muscle biopsy data.

For reasonable size studies such as this, the use of QUICKI and HOMA-IR are not gold standard for the measurement of insulin sensitivity. Additionally, during the fasting state these measurements are thought to mostly reflect hepatic insulin sensitivity. An insulin clamp would be ideal as an outcome measure, but a FSOGTT or OGTT with minimal modeling would also be fine.

The authors aren't really measuring fractional synthesis rate of muscle FFA because none of these FFA's are being synthesized from labeled precursors. And they do not appear to be measuring incorporation into the triglyceride pool, although from the reference cited it is a possibility. Therefore, it appears that the authors are mostly measuring intracellular FFA turnover rates – by looking at incorporation of the ¹³C lipid label into the muscle FFA pool while using the plasma TTR as the precursor pool from which these FFA are transported into muscle. If true, then the authors are not measuring the fractional synthesis of muscle FFA, and should change the manuscript to reflect that they are measuring intracellular turnover of FFA.

Minor:

Lines 59-59: "Better serum lipid profile" is subjective and not specific. It would be better to list what exactly changed rather than to say "better".

Line 99: The authors state that reference 9 showed decrease "desaturated" diacylglycerol content. However, the referenced study showed that exercise decreased "Di-saturated" diacylglycerol content – which is the opposite conclusion. However, a decrease of di-saturated diacylglycerol content agrees with the literature as well as the argument they authors are making regarding the negative impact of saturate lipid species. This and the argument relative to this conclusion should be changed. This should also be changes on line 267 where this data is also misrepresented.

Line 126: The athletes consumed more sugar – but based on the data provided it is speculation whether or not the sugar came from sports drinks.

Line 126: The energy intake data does not look correct. These data show that the athletes were only eating 1600kcal/day, yet were training on average 90 minutes per day which would be expected to burn 500-600kcal per hour (750-900kcal/session) given their VO₂peak values. Thus, the energy intake data are likely quite underestimated.

Lines 194-197: The authors are interpreting an increase in basal AKTser473 phosphorylation/total after exercise training as an increase in insulin signaling. However, this does not make sense because this was done in a basal biopsy – not an insulin stimulated biopsy which would be expected to increase AKT phosphorylation. It is difficult to interpret an increase in basal AKTser473 phosphorylation in a non-insulin stimulated biopsy without a change in basal plasma insulin concentration – but it certainly does not indicate increased insulin signaling.

Lines 243-234 – The conclusion that ectopic lipid accumulation could be a "new" target for health improvements in type 2 diabetes based on these data is not new. These types of conclusions have been published and discussed as a way to improve health for over 20 years.

Lines 303-304 – Again inappropriate discussion of improved insulin signaling in non-insulin stimulated biopsies.

Reviewer #2 (Remarks to the Author):

1. This study design with detraining of endurance-trained subjects and training of previously sedentary patients with type 2 diabetes provides novel and significant mechanistic information supporting the role for exercise as an important intervention to mitigate insulin resistance. The phenotypic outcomes assessed by 1H MRS, muscle biopsies and stable isotope turnover to comprehensively assess intramyocellular storage and turnover is a notable strength of the study.

2. The Results for 'Post exercise intervention comparisons of athletes and patients with type 2 diabetes' is not germane to the manuscript, i.e., the value in comparing detrained athletes vs trained T2D provides limited value. They intervention data are important to compare each of these groups in a pre-post design, but not to compare between groups after intervention. These data could be omitted.

3. Stating that differences in saturated/unsaturated fat in IMCL cannot be attributed to changes in nutritional intake may not be true, since accurate reliable assessment of nutritional intake was not performed. The authors should state this limitation.

4. Since deconditioning in the athletes resulted in no change in total, saturated or unsaturated triacylglycerols, diacylglycerols and ceramides or insulin signaling pathways, does this imply that the effects of training on insulin sensitivity are not associated with effects on IMCL?

5. The sex of the subjects should be indicated, since this could be a major confounding variable.

6. In comparisons in athletes vs T2D, the authors should discuss why none of these IMCL appear to differ between athletes and T2D, which is in contrast to many prior studies, including the original Athletes Paradox paper.

7. The authors need to discuss the primary novel results, i.e., IMCL turnover in context with the apparent negative findings of a lack of a difference or change in the IMCL species. While it is appreciated that IMCL species may not have differed because of subcellular localization etc., the authors state that these results are not surprising. Based on a considerable body of evidence to the contrary, it is indeed surprising that none of the IMCL species appeared to differ between groups or change with intervention. Since IMCL storage is in the title of the manuscript this deserves further consideration and discussion.

8. The authors do not adequately explain the fractional lipid mass determined by 1H MRS and how these results are to be interpreted.

9. How are the 1H MRS lipid content correlated with any of the biopsy-derived IMCL content? Are there concordances? Discrepancies?

10. It is not clear how intramyocellular storage and turnover might provide a new potential target for health improvement in type 2 diabetes patients. Clearly, exercise improved these parameters, but not clear how this could be a target for non-exercise interventions.

11. Much of the data in Table 1 should be moved to a Supplemental Table since much of these data are not primary to the manuscript.

Reviewer #3 (Remarks to the Author):

Referee report of the manuscript entitled: "Insulin Resistance in Type 2 Diabetes is Moderated by Saturated Fatty Acids Abundance and Turnover".

This manuscript reports the results of an experimental study aimed at describing the differences in intramyocellular lipids between insulin-sensitive athletes and insulin resistant patients with type 2 diabetes (T2D).

The manuscript addresses an important clinical question, it has an interesting experimental design and is well written; however, there are issues in the statistical analysis approach and interpretation of study results as described below.

1. The title should be more tentative for this exploratory study which was based on a modest sample of male participants.

2. Study design: It is stated on lines 101-102 pg. 5 that there is equipoise regarding the relative contributions of saturated and unsaturated intramyocellular fat to insulin sensitivity. Does this apply, or is it relevant to athletic individuals? Further clarity on why it was necessary to subject this group to deconditioning in this study would be helpful.

3. Study objectives and sample size calculation: Please specify what differences in saturated/unsaturated fatty acid content are of primary and secondary interest. Are they differences between athlete and T2D groups following the intervention, differences in changes with respect to baseline between those two groups, or something else? The sample size calculation described in the study protocol only alludes to assumptions on the fatty acids peaks suitable for healthy individuals but not patients with T2D, a single sample size calculation is reported, is this for saturated or unsaturated

fatty acid peaks? What is the assumed standard deviation for changes in fatty acid peaks with respect to baseline?

4. Statistical analysis:

4a. Baseline characteristics should be described using means and standard deviation (SD), not standard error of the mean (SEM). Do not include P-values for descriptive analyses, although it is common practice to report these in the literature, it is not good statistical practice to do so (refer to reporting guidelines for clinical trials and observational studies).

4b. The statistical analysis section should be revised. Please clarify the hypotheses or objectives of the study and ensure that each of those are addressed in the statistical analysis with the appropriate statistical analysis. For this exploratory study, reporting of confidence intervals for within- or between-group differences as opposed to means (SEM), parametric or non-parametric hypothesis tests would be appropriate. Revise Tables and Figures accordingly.

5. Results:

5a. Figure 1: Please add information on participant withdrawals to consort diagram.

5b. Table 1: Descriptive statistics of the study sample should be based on means (SD) and P-values removed as described above.

5c. Pg. 5 line 137: Replace "Table 2" by "Table 1".

5d. There seems to be a discrepancy in the reporting of baseline weight between Tables 1 and 2.

**Subject: NCOMMS-23-12748-R1 - Intramyocellular Saturated Fatty Acid
Abundance and Turnover - an Index of Exercise-driven Metabolic Health**

20th September 2023

We would like to thank the editors and reviewers for their insightful comments. These are responded to in full and have substantially improved the manuscript. In this letter, our responses are in normal font. In the revised manuscript all additions/changes to the text are highlighted in yellow. Please note that all Tables and Figures have been re-drawn as requested.

REVIEWER COMMENTS

Reviewer #1 (Remarks to the Author):

The authors performed a study to evaluate differences in muscle lipid metabolism and storage to help explain the “athlete’s paradox” in male athletes and individuals with type 2 diabetes. The focus of the study is alterations in muscle triglyceride storage, triglyceride saturation, muscle lipid accumulation, plasma and muscle FFA turnover, and insulin sensitivity. The studied athletes before and after 4 weeks of detraining, and individuals with type 2 diabetes before and after 8 weeks of endurance training. The authors found that athletes have greater storage of saturate triacylglycerol by MRI but not muscle biopsy analysis with greater palmitate kinetics that is dampened by detraining. Individuals with type 2 diabetes have more muscle triglyceride that is more unsaturated and blunted FFA kinetics, both of which are improved after training.

Major: The authors appear to have measured insulin signaling using non-insulin stimulated muscle biopsies. Therefore, these insulin signaling data are not appropriate as no differences in insulin signaling would be expected since they are only measuring basal signaling. Additionally, the specific phosphorylation sites listed in table 4 should be stated in the table, rather than forcing the reader to read supplemental methods because these phosphorylation sites are key to understanding the data shown in Table 4. For instance, for IRS1 – did the authors measure serine phosphorylation (inhibitory) or tyrosine phosphorylation (stimulatory)? This should be made more obvious to the reader. Further, the authors are trying to make conclusions on insulin signaling based on basal non-insulin stimulated muscle biopsies – which is not how these measurements are performed. To make conclusions about insulin signaling the authors would need to compare the increase in insulin signaling measurements (for instance the AKTser473 phosphorylation/total) in basal compared to insulin stimulated conditions before and after the intervention. Just comparing basal measurements of insulin stimulated insulin signaling is not appropriate for these conclusions.

Thank you for raising the importance of basal *versus* insulin-stimulated insulin signalling targets' assessment. First, it is crucial to emphasize that the principal aim of our work was to investigate intramyocellular lipid structure (MRI spectroscopy) and turnover (stable isotope studies combined with biopsies). Insulin stimulation (a hyperinsulinaemic-euglycaemic clamp) is not required for these aims. Molecular targets of skeletal muscle insulin signalling and whole body insulin sensitivity were included in the investigation for a comprehensive assessment of changes resulting from the exercise interventions but were not in themselves hypotheses, aims, objectives or part of any sample size calculations. We agree with the reviewer that in an ideal scenario both these investigations should be performed in basal to insulin stimulated assessments.

At the inception of the study, we had considered a hyperinsulinaemic-euglycaemic clamp as part of the experimental protocol. It is important to fully explain why in the end, as a pragmatic approach, this was not included:

- a. The duration for each study visit was 7.5 hours: 90 min cardiac MRI and skeletal muscle spectroscopy, 45 min for bloods, set up and execution of the CPEX, a 4-hour stable isotope infusion with additional time needed for pre and post infusion biopsies – for all these investigations the participants had to be *fasted*. They were followed by inspection of physical exercise questionnaires and food diaries, accelerometer data transfer (45 min), whilst participants had refreshments. Clinical safety also required that all patients have a full set of blood results (FBC, u&e and LFT) and clinical assessment before they left the unit, due to the amount of albumin infused intravenously for 4 hours as part of the fatty acid isotope complexing in solutions (3:1 ratio with albumin), which attracts a small risk of pulmonary oedema, particularly in diabetes patients.
- b. If done, addition of a clamp should have overlapped with both spectroscopy and isotope infusions, as otherwise structure and turnover would have been assessed in different states. Performing a clamp in a patient inside the MRI scanner and subsequently transferring a patient from the MRI suite to the clinical research facility during a clamp raises important clinical risks. Even if performed only for turnover (during pre-infusion biopsy, 4 hour isotope infusions and post-infusion biopsy) – this would have taken at least 5 hours, to which an initial period of clamp stabilisation would have also been required. For all these practical reasons, time constraints, and clinical risk the option of a clamp was removed from the protocol. The option of a clamp was raised as a concern by both the Ethics Committee and the reviewers of the grant application at the British Heart Foundation, particularly that a clamp was not necessary to assess intramyocellular lipid structure or turnover, which were the primary aim of our investigation.
- c. Additionally, in order to assess basal level of phosphorylation *versus* insulin-stimulated phosphorylation of key targets post-Insulin Receptor signalling, the number of skeletal muscle biopsies should have been doubled from 2 per visit to 4 per visit, increasing the risk of bleeding, infection, local trauma and prolonged healing. Indeed, this is easily achievable in mouse studies and cell lines where treatments of pre- and post- are possible and ethical. In our study, the aim was to assess the potential medium-term effect of exercise interventions (deconditioning *versus* exercise training) in healthy and type 2

diabetes human participants. For example, it is well established that exercise stimulates phosphorylation of AMPK α (Thr 172 site) and that this would be expected to change. Therefore, we chose to specifically test this target. Likewise, mTOR signalling affects phosphorylation of IRS-1 at Ser612 site (in humans) which is the specific site we tested. Indeed, there are over 70 putative Ser/Thr and 20 Tyr phosphorylation sites on IRS1 and in some cases these are inhibitory and some activatory. However, not all serine IRS1 phosphorylation is inhibitory (as shown by Morris White in [https://www.cell.com/cell-metabolism/fulltext/S1550-4131\(09\)00340-4?script=true](https://www.cell.com/cell-metabolism/fulltext/S1550-4131(09)00340-4?script=true); we assume that the reviewer was referring to the S307 site which a decade ago was hypothesized to be inhibitory; however in vivo studies using knockout of iS307 IRS1 in mice revealed that the S307 IRS1 KI mice exhibited more severe insulin resistance than controls therefore suggesting a more complicated relationship between in vitro and in vivo findings. We chose the Ser 612 site as our specific target site on IRS1, as it was relevant to the intervention investigated in our human participants, which is the site reported to be regulated by the TORC complex. Furthermore insulin-independent (heterologous) kinases have been shown to be able to phosphorylate IRS1/2 under basal conditions (e.g AMPK) (<https://www.ncbi.nlm.nih.gov/pmc/articles/PMC4011499/>).

- d. Finally, a clamp may have influenced the lipid turnover in a manner that would have been difficult to anticipate for including in our sample size calculations or final data interpretation. We are not aware of any study in humans who have assessed stable isotope *lipid* turnover during an insulin clamp and there remains a concern that it may have confounded the results of the turnover.

It would not be possible to repeat the study for just the purposes of a hyperinsulinaemic euglycaemic clamp: the participants to the original study will not have the same cardiometabolic health status (some athletes may no longer be competitive recreational athletes as people move on in their lives, and we know that some of the diabetes participants have developed myocardial infarctions and were admitted through our Cardiology department). Recruiting a new set of participants would require new funding, new fellow, new Ethics as a separate study and cannot be achieved in the time remit of a manuscript rebuttal. Most likely different sample sizes may be needed due to the influence of the clamp on the fractional incorporation rate of saturated/unsaturated lipids, for which we are not aware of any informative data.

We do appreciate the points raised by the reviewer and we added the following in the manuscript:

- Clarified throughout that we only report basal status of insulin receptor pathway targets (removed any reference to “signalling”), in situ, in human participants, exploring only if deconditioning/exercise training alone induces changes in their basal, non-stimulated state (all Results sections, Discussion and Tables 3 and 4 titles). It is however quite striking that exercise training alone in patients with type 2 diabetes significantly increases basal phosphorylation of AMPK α (T172) and Akt (S473), which we mentioned in Discussion, page 14.
- We added the specific sites of IRS-1Ser612, pAKT Ser 473 and pAMPK T172 to Table 4, as requested.

- We added in study limitations the main 3 pragmatic reasons why the hyperinsulinaemic-euglycaemic clamp was not included in the protocol (lines 335-6).

The lipidomic data is concerning because it does not appear that the methods used had representative saturated and unsaturated ceramide, diacylglycerol, and triglyceride standards used to quantify saturated and unsaturated lipid species. They only appear to have saturated internal standards. Therefore, the authors are assuming that ionization and calculated abundances of both saturated and unsaturated species are identical – which is not always true and could be driving differences in the data. Ideally, the lipidomic analysis would have representative standards against which the ratio of analyte to internal standard could be quantified that would be specific for each species. Interpretation is difficult as it does not appear that the lipidomic method has been published.

The reviewer is correct that there are differences in ionisation efficiency between individual ceramides, and degree of unsaturation makes a difference. However, it has been a difficulty getting labelled standards for a diverse range of lipids. For our method we would describe it as semi-quantitative which measures *relative* rather than absolute changes. The experimental approach taken in our study is consistent with protocols that are routinely utilised as part of lipidomic analyses. This can be evidenced through multiple publications in leading journals and methodological reference books e.g. Kurzawa-Akanbi et al. Acta Neuropathol. 2021; 142:961-984; Suchacki et al. eLife 2023; 12: e88080; Thompson et al Sci. Rep. 2023; 13: 3937. [published by co-author PW] or those of others, such as Kindt et al. Nat. Commun. 2018; 9: 3760; Mundra et al. JCI Insight 2018; 3: e121326; Drotleff and Lämmerhofer. Anal. Chem. 2019; 91: 9836-9843; Munjoma et al. J. Proteome Res. 2022; 21: 2596-2608; Zhang et al. J. Lipid Res. 2022:100218; Huynh et al. Methods Mol Biol. 2023; 2628: 489-504.

For structural quantification we would need a range of labelled ceramides. Were this work to be taken forward in a bigger screening study we would opt for a targeted approach where such labels could be included and focus on a limited number of lipid species. We have included this limitation in the revised manuscript, lines 336-7.

There is internal inconsistency in this manuscript that needs to be addressed. There was a significant change in saturated and unsaturated TAG storage by MRI, but not by biopsy, after the exercise training intervention with individuals with type 2 diabetes. Therefore it is difficult to understand which data should be believed. The current discussion of this topic should be expanded. Further, the muscle biopsy TAG data should be calculated as percent saturation/unsaturation so that these data can be compared to the MRI data. Currently it is difficult to compare these two measurements. For instance, lines 258-259 appears to be describing differences in the TAG data by MRI, as there did not appear to be differences in saturated TAG storage from the muscle biopsy data.

We thank Reviewer 1 for raising such an important point of an *apparent* inconsistency between the MR spectroscopy and the muscle lipidomic data. We are

pleased to reassure the Editors and Reviewers that there is no inconsistency in the data presented. The two methods assess saturation in two rather different ways:

- a. The MR spectroscopy measures the proportion of saturated and unsaturated carbon bonds present in the intramyocellular lipid stores. Thus, all the saturated carbon bonds contained in both saturated and unsaturated fatty acids are included in the fraction of saturated intramyocellular lipid – this is why, by MR spectroscopy this saturated component represents $\approx 85\%$ of the whole intramyocellular lipid store. In fact, the actual proportion of intramyocellular saturated *fatty acids per se* is much smaller – see below – which is completely understandable, as for any given carbon chain length there is only one saturated fatty acid and multiple unsaturated ones. Further, and of note, is that the spectroscopic MR signal is acquired from only the intramyocellular lipid stores, as MR spectroscopy does not have the signal-to-noise ratio to receive information from lipids contained in cell membranes for example.
- b. In contradistinction, in the case of the muscle lipidomic analysis, the results present a proportional representation of the sum of saturated and unsaturated fatty acids. Importantly, they are contained in the whole muscle – ie the intramyocellular lipid stores + the cell membranes + the extramyocellular compartment found in between myocytes (which cannot be dissected out). In the muscle lipidomic analysis, total triglycerides represent the majority of species $\approx 97.8\%$, (with diacylglycerols $\approx 2\%$ and ceramides $\approx 0.2\%$). Of these, only $\approx 1.1\%$ of triglycerides, $\approx 48\%$ of diacylglycerols and $\approx 71\%$ of ceramides are saturated (representative percentages for the raw data presented in Table 3).

Hopefully this better illustrates that the two techniques assess lipid saturation from completely different angles and they cannot be compared side by side, thus presentation by percentages of saturated/unsaturated lipid species would not be advantageous.

Since we explained that MR spectroscopy data shown only pertains to the intramyocellular lipid storage, we now add the MR acquisition of the extramyocellular lipid peaks which are detected at 1.1 and 1.5ppm (saturated) and 2.3 and 5.5 ppm (unsaturated). In Figures 1-2 Supplemental Material we show that there were no differences in the extramyocellular lipids within or between groups before or after exercise interventions. Further, we present a separate voxel acquisition from adipose tissue and that there were no within or between groups differences in the adipose tissue either. This strongly suggests that the extramyocellular and the adipose tissue compartments are similar between subjects, and not beneficially related to these exercise interventions. This suggests that the intramyocellular lipid storage is the metabolically active compartment of interest. This intramyocellular compartment can be serially and non-invasively interrogated with the MR spectroscopy technique.

In order to better reflect these concepts, we have made the following changes to the manuscript:

- We added the terms “saturated/unsaturated carbon *bonds* within stored lipids” in the Intro, Methods, all Results, Figure 2 title/legend, to express more specifically what spectroscopy measures. Additionally, we included a more detailed explanation in Discussion about the MR spectroscopy and muscle

lipidomics assessment in particular what their respective data depict (lines 269-277)

- We added the extramyocellular and adipose tissue MR spectroscopy data to the Results (lines 226-229).and the Supplemental Figures 1-2 are included in the Supplemental material

For reasonable size studies such as this, the use of QUICKI and HOMA-IR are not gold standard for the measurement of insulin sensitivity. Additionally, during the fasting state these measurements are thought to mostly reflect hepatic insulin sensitivity. An insulin clamp would be ideal as an outcome measure, but a FSOGTT or OGTT with minimal modeling would also be fine.

We thank the reviewer for this insightful comment and agree with their view: the fasting state is more reflective of the basal insulin sensitivity of the whole body, primarily hepatic. As mentioned above, we did consider a hyperinsulinaemic euglycaemic clamp in our initial study design but was pragmatically removed following the peer review at the Project Grants Committee of the British Heart due to the added risk, duration and complexity which were not directly beneficial to the primary aim/hypothesis of the study, of a longitudinal exercise intervention effect on intramyocellular lipid abundance and turnover.

An OGTT could have been performed only at the end of the protocol investigations which required participants to remain fasted, specifically after the intravenous isotope infusions. However this would have extended the 7.5 hour study visit by another 2-3 hours.

It is well known that physical activity is an important determinant of whole body and skeletal muscle insulin sensitivity. In our study we included QUICKI and HOMA-IR as a demonstrable measure of whole-body insulin sensitivity *change* post exercise interventions rather than a cross-sectional skeletal muscle insulin sensitivity assessment.

Insulin sensitivity or insulin signalling or pancreatic beta-cell function were not the principal aims of the study. We are acceptant that this is a compromise within the constraints explained and mentioned this in the study limitations. We have added that only basal, non-stimulated insulin sensitivity has been assessed, throughout the manuscript.

The authors aren't really measuring fractional synthesis rate of muscle FFA because none of these FFA's are being synthesized from labeled precursors. And they do not appear to be measuring incorporation into the triglyceride pool, although from the reference cited it is a possibility. Therefore, it appears that the authors are mostly measuring intracellular FFA turnover rates – by looking at incorporation of the 13C lipid label into the muscle FFA pool while using the plasma TTR as the precursor pool from which these FFA are transported into muscle. If true, then the authors are not measuring the fractional synthesis of muscle FFA, and should change the manuscript to reflect that they are measuring intracellular turnover of FFA.

We entirely agree with the reviewer regarding the nomenclature used and their interpretation of our intramyocellular turnover is correct. This is our first foray in

this field of work and we used the most common nomenclature cited in literature, not wishing to introduce new terminology and risk being criticised for this. Indeed, there is no neo-lipogenesis and the “synthesis rate” is a misnomer. We are very pleased to change this throughout the manuscript to Fractional incorporation rate (FIR), which we also feel is better reflective of the intramyocellular turnover process measured.

Minor:

Lines 59-59: “Better serum lipid profile” is subjective and not specific. It would be better to list what exactly changed rather than to say “better”.

Thank you, serum lipid profile has not been replaced specifically with “serum cholesterol/triglycerides”

Line 99: The authors state that reference 9 showed decrease “desaturated” diacylglycerol content. However, the referenced study showed that exercise decreased “Di-saturated” diacylglycerol content – which is the opposite conclusion. However, a decrease of di-saturated diacylglycerol content agrees with the literature as well as the argument they authors are making regarding the negative impact of saturate lipid species. This and the argument relative to this conclusion should be changed. This should also be changes on line 267 where this data is also misrepresented.

Thank you for highlighting this very important point. We removed the misplaced reference 9 and added additional supportive evidence that saturated and unsaturated fatty acids have different metabolic fates within skeletal muscle (<https://pubmed.ncbi.nlm.nih.gov/22796147/>) and that SIRT1 protein and activity in skeletal muscle was increased by a high-fat diet enriched in saturated fatty acids in humans, but was not affected by a high-fat diet enriched in unsaturated fatty acids (<https://pubmed.ncbi.nlm.nih.gov/30707625/>). These were included only in the Introduction (lines 96-101) but removed from the Discussion to avoid repetition.

Line 126: The athletes consumed more sugar – but based on the data provided it is speculation whether or not the sugar came from sports drinks.

Thank you for raising this point. We agree this appears speculative as presented at baseline characteristics (Table 1), and removed this comment from (now) line 130.

Line 126: The energy intake data does not look correct. These data show that the athletes were only eating 1600kcal/day, yet were training on average 90 minutes per day which would be expected to burn 500-600kcal per hour (750-900kcal/session) given their VO₂peak values. Thus, the energy intake data are likely quite underestimated.

We acknowledge that food diaries are both subjective and an inexact measure of intake. Under-reporting is well recognised (<https://doi.org/10.1079/BJN2000281>), so we included this in our discussion (lines 292-4).

As this was not a nutrition study/intervention, we did not provide the actual meals to participants so the reason we asked participants to record food diaries was two-fold:

- a. to ensure they had broadly comparable diets and

- b. to ensure that food intake did not change significantly for the duration of the study (ie someone who may have decided to go on a specific diet during this time).

Lines 194-197: The authors are interpreting an increase in basal AKTser473 phosphorylation/total after exercise training as an increase in insulin signaling. However, this does not make sense because this was done in a basal biopsy – not an insulin stimulated biopsy which would be expected to increase AKT phosphorylation. It is difficult to interpret an increase in basal AKTser473 phosphorylation in a non-insulin stimulated biopsy without a change in basal plasma insulin concentration – but it certainly does not indicate increased insulin signaling.

The reviewer is correct that this is not an increase in insulin-stimulated phosphorylation of Akt but instead, of the basal phosphorylation status post-exercise intervention; we have therefore amended the wording to reflect this. Specifically, we state that we detect these changes post exercise intervention, there is increased basal phosphorylation of Akt and AMPK, which may serve as indicators of enhanced post insulin receptor pathway and metabolic sensing. The latter terminology replaces all previous reference to “insulin signalling” in the revised manuscript.

Lines 243-234 – The conclusion that ectopic lipid accumulation could be a “new” target for health improvements in type 2 diabetes based on these data is not new. These types of conclusions have been published and discussed as a way to improve health for over 20 years.

Thank you for raising this point. We agree that the concept of “ectopic intramyocellular lipid accumulation” is not new. However, what we demonstrated in this work, is that the *saturated* intramyocellular lipid accumulation and turnover tracks with the increased athletic performance in healthy subjects and better all-around health markers in type 2 diabetes patients. This finding is novel and important and contradicts the dogma that “saturated fat is bad for you”, showing that at intramyocellular level, saturated fat has a distinctive role, previously unappreciated. We have emphasized this in Discussion, lines 251-255, 265-267, and 296.

Lines 303-304 – Again inappropriate discussion of improved insulin signaling in non-insulin stimulated biopsies.

Thank you, as above, the mention of basal state (non-insulin stimulated) is now present throughout the manuscript and references to insulin signalling have been removed and replaced with “insulin receptor pathway” throughout the manuscript.

Reviewer #2 (Remarks to the Author):

1. This study design with detraining of endurance-trained subjects and training of previously sedentary patients with type 2 diabetes provides novel and significant mechanistic information supporting the role for exercise as an important intervention to mitigate insulin resistance. The

phenotypic outcomes assessed by 1H MRS, muscle biopsies and stable isotope turnover to comprehensively assess intramyocellular storage and turnover is a notable strength of the study.

We thank Reviewer 2 for their praising remark of our complex mechanistic experimental medicine study.

2. The Results for ‘Post exercise intervention comparisons of athletes and patients with type 2 diabetes’ is not germane to the manuscript, i.e., the value in comparing detrained athletes vs trained T2D provides limited value. They intervention data are important to compare each of these groups in a pre-post design, but not to compare between groups after intervention. These data could be omitted.

We thank Reviewer 2 for this observation. The detrained athletes are the surrogate equivalent of matched healthy (untrained) volunteers for the type 2 diabetes patients. Whilst the pre/post intervention intra-group comparator shows that each group has changed as a result of this intervention, it is only when the type 2 diabetes group are cross-compared with their “matched, detrained healthy volunteer” counterparts that one is able to appreciate how much beneficial change and how many of the adverse features seen at baseline in the type 2 diabetes phenotype were reversed by an 8-week exercise program. At Reviewer 3 request, we added a different way of presenting the data, as differences of the mean (CI) within-group changes – which effectively contains the post-intervention between-groups comparator data.

3. Stating that differences in saturated/unsaturated fat in IMCL cannot be attributed to changes in nutritional intake may not be true, since accurate reliable assessment of nutritional intake was not performed. The authors should state this limitation.

We thank Reviewer 2 for this observation. As already mentioned to R1 above, food diaries are both subjective and an inexact measure of intake. Indeed, as we have not provided the actual meals to participants (as a nutritional intervention study would), we cannot exclude an influence from nutritional intake, and have acknowledged this in the study limitations (lines 339-340).

4. Since deconditioning in the athletes resulted in no change in total, saturated or unsaturated triacylglycerols, diacylglycerols and ceramides or insulin signaling pathways, does this imply that the effects of training on insulin sensitivity are not associated with effects on IMCL?

We thank Reviewer 2 for raising this important point. We did not really expect that 4 weeks of deconditioning would alter the intramyocellular lipid composition (abundance of total, saturated/unsaturated fraction) in a group of “super-healthy” individuals. Although the study was not powered to look at the differences with deconditioning in athletes group, we observed that the *saturated* intramyocellular lipids (palmitate) turnover decreased proportionately much more than the linoleate post-deconditioning – trending statistical significance ($p=0.07$). In context with the rest of the study findings, this is in keeping with the concept that the saturated intramyocellular lipid compartment is the one responsible for athletic performance, most likely being the preferential substrate, as it requires least energy for beta-oxidation. This concept is referred to in the discussion, paragraph starting with line 264.

5. The sex of the subjects should be indicated, since this could be a major confounding variable.

Thank you for raising this point – we agree entirely and added this to the abstract as per journal instructions. It is also specified in the Methods (line 369) and the reason for recruiting male participants only in this first of its kind study is explained in the Discussion (line 257).

6. In comparisons in athletes vs T2D, the authors should discuss why none of these IMCL appear to differ between athletes and T2D, which is in contrast to many prior studies, including the original Athletes Paradox paper.

We thank Reviewer 2 for raising an important observation. Importantly, the *total* IMCL in type 2 diabetes and trained athletes reported by us with magnetic resonance spectroscopy is identical to that described by Dr Goodpaster in the original athletes paradox paper using quantitative histochemical analysis of biopsy specimens (<https://pubmed.ncbi.nlm.nih.gov/11739435/>). The contribution of intramyocellular lipids to insulin sensitivity is complex and has been extensively studied, with investigators reporting diverse results. As eloquently reviewed elsewhere (<https://doi.org/10.2337/dbi18-0042>) it has increasingly become apparent, from multiple studies, that it would be simplistic to expect that the total amount or concentration of triglycerides, diacylglycerols or other species (sphingolipids, acylcarnitines) would immediately explain changes between insulin sensitive/resistant subjects or a longitudinal change within the same group of patients with metabolic syndrome. Notwithstanding conflicting reports of such total changes, it has become apparent that the matter is more refined and specific intramyocellular lipid species (or even specific isoforms) may be responsible (<https://doi.org/10.1007/s00125-015-3850-y>, <https://doi.org/10.1016/j.cmet.2014.09.015>) or their subcellular localisation (<https://doi.org/10.1007/s00125-011-2419-7>, DOI: [10.1172/jci.insight.96805](https://doi.org/10.1172/jci.insight.96805)), for example their proximity to the mitochondria. These have been added to the Discussion, lines 300-303 and 305-310.

7. The authors need to discuss the primary novel results, i.e., IMCL turnover in context with the apparent negative findings of a lack of a difference or change in the IMCL species. While it is appreciated that IMCL species may not have differed because of subcellular localization etc., the authors state that these results are not surprising. Based on a considerable body of evidence to the contrary, it is indeed surprising that none of the IMCL species appeared to differ between groups or change with intervention. Since IMCL storage is in the title of the manuscript this deserves further consideration and discussion.

We thank Reviewer 2 for raising this and acknowledge the complexity of the field. We removed the “not surprising” and contextualised the findings carefully, as results remain at variance (lines 305-310).

8. The authors do not adequately explain the fractional lipid mass determined by 1H MRS and how these results are to be interpreted.

The fractional lipid mass (normalised to water) was calculated from the individual spectral contributions of the main lipid peaks detected from the intramyocellular lipid storage: 0.9 ppm and 1.3 ppm (both saturated) and the 2.1 ppm plus the olefinic peaks at 5.2/5.3 ppm (unsaturated). This was re-written in Methods (lines 414-15). We have already explained above, to Reviewer 1, the concept of detecting saturated vs unsaturated carbon bonds within all lipids (ie all saturated carbon bonds contained in both saturated and unsaturated fatty acids are therefore included in the calculation of fraction of saturated lipids in the magnetic resonance spectroscopy investigation.

9. How are the ¹H MRS lipid content correlated with any of the biopsy-derived IMCL content? Are there concordances? Discrepancies?

Thank you for raising this point, which follows from the previous question, and this has also been extensively explained at responses provided to Reviewer 1. The magnetic resonance spectroscopy evaluation of saturated/unsaturated compartments is significantly different than the classical, proportional representation of the lipidomic analysis which detects each fatty acid individually. Magnetic resonance spectroscopy cannot detect individual fatty acids and their metabolism is too slow for attempting any *in vivo* hyperpolarised tracers for example (which have a half-life of 60-80 seconds requiring almost instantaneous imaging after bolus administration). Magnetic resonance spectroscopy detects different precession frequencies of protons contained in saturated or unsaturated carbon bonds, as shown in Figure 2A. Therefore it provides an overall evaluation of relative proportions of saturated/unsaturated carbon bonds, without ability to assign them to individual fatty acids. Since the saturated carbon bonds are contained not only in saturated fatty acids, but in all of the unsaturated fatty acids, in various amounts, it would be questionable what a direct comparison or correlation between these may mean, particularly that spectroscopy does not provide the information on whether they are contained in TAG, DAG or ceramides. We have ran these correlations, and below we present the Pearson's correlation analysis of the ¹H-MRS saturated/unsaturated intramyocellular lipids with the saturated/unsaturated TAG, DAG and ceramides. Whist some correlation or trends can be seen, we believe it may be difficult to draw any firm conclusion from these data.

	Saturated IMCL in Athletes at Baseline		Saturated IMCL in Type 2 Diabetes Patients at Baseline		Saturated IMCL in Athletes after Deconditioning		Saturated IMCL in Type 2 Diabetes patients after exercise training			Unsaturated IMCL in Athletes at Baseline		Unsaturated IMCL in Type 2 Diabetes Patients at Baseline		Unsaturated IMCL in Athletes after Deconditioning		Unsaturated IMCL in Type 2 Diabetes patients after exercise training	
	r	p-value	r	p-value	r	p-value	r	p-value		r	p-value	r	p-value	r	p-value	r	p-value
Lipidomics analyses																	
Saturated TAG at baseline	0.04	0.92	-0.49	0.18					Unsaturated TAG at baseline	0.03	0.93	0.47	0.20				
Saturated DAG at baseline	0.04	0.93	-0.62	0.07					Unsaturated DAG at baseline	-0.50	0.17	0.72	0.03				
Saturated Ceramides at baseline	-0.35	0.36	-0.63	0.07					Unsaturated Ceramides at baseline	0.51	0.16	0.51	0.16				
Saturated TAG after exercise intervention					0.39	0.27	0.16	0.66	Unsaturated TAG after exercise intervention					-0.57	0.09	-0.16	0.67
Saturated DAG after exercise intervention					0.22	0.54	-0.52	0.12	Unsaturated DAG after exercise intervention					-0.18	0.63	0.76	0.01
Saturated Ceramides after exercise intervention					0.49	0.15	-0.04	0.92	Unsaturated Ceramides after exercise intervention					-0.21	0.56	0.06	0.87
r = Pearson's correlation coefficient IMCL = intramyocellular lipids TAG = triacylglycerol DAG = diacylglycerol																	

10. It is not clear how intramyocellular storage and turnover might provide a new potential target for health improvement in type 2 diabetes patients. Clearly, exercise improved these parameters, but not clear how this could be a target for non-exercise interventions.

Thank you for raising this. In order to measure any change as a result of a pharmacological or non-pharmacological intervention, it is important to have a *biomarker* that is easy to measure. At its first implementation, the ¹H-magnetic resonance spectroscopy study was taking 1 hour to acquire. We have since refined the sequence significantly, and additionally, adapted it to the moving heart. Currently, the sequence takes 4 min in skeletal muscle and 6 min in the heart. This offers a tremendous platform for investigating the intramyocellular lipid pool as a potential future target for health interventions in diabetes. As Cardiologists, we are also interested in the myocardium, as heart health has an important survival prognostic influence in type 2 diabetes patients. It is important to mention that despite many well conducted trials of glucose lowering therapies the overall longterm cardiovascular survival outcome of type 2 diabetes patients has improved little. The ectopic intramyocellular fat accumulation which is intimately linked to the development of insulin resistance has received little attention as a potential modulator of cardio-metabolic health, even though it is increasingly recognised as the primary or initiating defect that is evident decades before clinical diabetes becomes apparent. Having a biomarker to measure this non-invasively could be a step change in future research as well as clinical patient care. For example, an immediate non-exercise intervention could be to test the re-purposing of fenofibrate, a Peroxisome Proliferator Activated Receptor alpha (PPAR α) agonist, which augments intramyocellular fatty acid oxidation as well as reducing serum triglycerides. Fenofibrate has been shown to reduce both myocardial lipid content and fibrosis in a rat experimental model of diabetes (*Cardiovasc Diabetol.* 2009; 8: 16-16), and demonstrated a significant risk reduction in both major adverse cardiovascular events and hospitalisations for heart failure in type 2 diabetes patients, particularly those with significant dyslipidaemia (*Diabetes Care.* 2022; 45: 1584-1591, *Diabetes Care.* 2022; 45: 1500-1502). We currently have a grant in review where we propose to carry out this investigation, examining both the skeletal muscle and myocardial lipid structure and turnover, as a result of fenofibrate therapy. In this investigation we will establish if the non-exercise intervention (PPAR α agonist) has comparable effects on skeletal (and cardiac) muscle intramyocellular lipids as exercise training does.

11. Much of the data in Table 1 should be moved to a Supplemental Table since much of these data are not primary to the manuscript.

Thank you for this suggestion, Table 1 of the baseline characteristics of the participants has now been moved to the Supplement.

Reviewer #3 (Remarks to the Author):

Referee report of the manuscript entitled: "Insulin Resistance in Type 2

Diabetes is Moderated by Saturated Fatty Acids Abundance and Turnover”.

This manuscript reports the results of an experimental study aimed at describing the differences in intramyocellular lipids between insulin-sensitive athletes and insulin resistant patients with type 2 diabetes (T2D).

The manuscript addresses an important clinical question, it has an interesting experimental design and is well written; however, there are issues in the statistical analysis approach and interpretation of study results as described below.

We thank Reviewer 3 for their praising comments on our mechanistic experimental study.

1. The title should be more tentative for this exploratory study which was based on a modest sample of male participants.

We acknowledge this and have changed the title accordingly, to “Intramyocellular Saturated Fatty Acid Abundance and Turnover - an Index of Exercise-driven Metabolic Health”.

2. Study design: It is stated on lines 101-102 pg. 5 that there is equipoise regarding the relative contributions of saturated and unsaturated intramyocellular fat to insulin sensitivity. Does this apply, or is it relevant to athletic individuals? Further clarity on why it was necessary to subject this group to deconditioning in this study would be helpful.

Thank you for this important point. The contributions to insulin sensitivity of saturated/unsaturated intramyocellular lipids in athletes were of interest but were not the central hypothesis. There is evidence that in healthy people (with or without family history of type 2 diabetes), it takes only 9 days of being sedentary in order to have a significant, measurable increase in *basal* insulin resistance (HOMA-IR) (<https://doi.org/10.2337/db09-0369>). Since multiple transcriptional changes relevant to skeletal muscle metabolism were observed in these healthy individuals after 9 days of bed rest (doi: 10.1152/ajpendo.00590.2009) we considered important that the athletic control group is studied in both trained and deconditioned status, to explore if any changes in abundance or turnover of intramyocellular lipid saturation accompany the change in basal insulin sensitivity. The 8-week exercise training of type 2 diabetes patients resulted in re-adaptations of their intramyocellular lipid abundance and turnover that aligned their phenotype with the deconditioned athlete (but not with the trained athlete). This is mentioned in the Discussion, lines 240-241. As skeletal muscle trained status can be quite rapidly lost through inactivity the study design was easier achieved by enrolling trained athletes rather than enrolling non-trained healthy people and working with them through an intensive period of training (resulting in drop-outs).

3. Study objectives and sample size calculation: Please specify what differences in saturated/unsaturated fatty acid content are of primary and secondary interest. Are they differences between athlete and T2D groups following the intervention, differences in changes with respect to baseline between those two groups, or something else? The sample size calculation

described in the study protocol only alludes to assumptions on the fatty acids peaks suitable for healthy individuals but not patients with T2D, a single sample size calculation is reported, is this for saturated or unsaturated fatty acid peaks? What is the assumed standard deviation for changes in fatty acid peaks with respect to baseline?

Thank you for asking us to clarify these important points. First, as explained in the methods, the fraction of saturated intramyocellular lipid is 1 minus the fraction of unsaturated intramyocellular lipid, so the changes track together and have the same SD for purposes of sample size calculations. The pilot data we based our sample size calculations was from young (20-25 years old) both male and female healthy, non-athletic individuals, with saturated/unsaturated fractions of $89.3 \pm 4.3\%$ and $10.7 \pm 4.3\%$, respectively (in protocol expressed in mM/l as in previous co-applicant Henning work at ETH in Zurich, DOI:[10.3929/ethz-a-009759787](https://doi.org/10.3929/ethz-a-009759787)). Our study participants were older, in their 5th decade of age, in order to age match with the type 2 diabetes patients and in addition were all male, to remove intra-group variability (intramyocellular lipid stores and utilisation rates are both higher in women, DOI: [10.1152/ajpendo.00266.2001](https://doi.org/10.1152/ajpendo.00266.2001)). We did not have preliminary data in type 2 diabetes patients, and the study hypothesis was that their saturated skeletal muscle abundance was different by an order of 10% compared to healthy status. We did not specify if these were healthy *athletic trained* or healthy *deconditioned* as we did not have that granularity of information – this information has been delivered through our study.

4. Statistical analysis:

4a. Baseline characteristics should be described using means and standard deviation (SD), not standard error of the mean (SEM). Do not include P-values for descriptive analyses, although it is common practice to report these in the literature, it is not good statistical practice to do so (refer to reporting guidelines for clinical trials and observational studies).

The SEM have been replaced by SD in (now) Supplemental Table 1. The *p* values for the descriptive analyses of past medical history and medications have been removed from Supplemental Table 1.

4b. The statistical analysis section should be revised. Please clarify the hypotheses or objectives of the study and ensure that each of those are addressed in the statistical analysis with the appropriate statistical analysis. For this exploratory study, reporting of confidence intervals for within- or between-group differences as opposed to means (SEM), parametric or non-parametric hypothesis tests would be appropriate. Revise Tables and Figures accordingly.

Thank you for these two important points. The hypotheses of the study were: 1) the relative proportions of saturated/unsaturated intramyocellular lipid storage in patients with type 2 diabetes is different to age/sex matched healthy athletes, and 2) exercise training in type 2 diabetes patients is associated with changes in the saturated/unsaturated intramyocellular storage and turnover. These have been added at the end of the introduction, lines 114-118.

As presented at point 3 above, we powered the study to observe a 10% difference for storage between diabetes and healthy status (either trained or deconditioned) and a 10% difference for storage after exercise training in the type

2 diabetes group. From our own data shown above at point 3, of $89.3 \pm 4.3\%$ saturated intramyocellular lipid in healthy, non-athletic trained young subjects we assumed the following: a change of population mean of 10% (8.93%), a SD for athletes of 5 (considering variation in training) and a higher SD (of 14) in type 2 diabetes patients (different duration of disease, metabolism), a study power of 0.8 and significance level of 0.05. The required sample size of 25 per group for detecting between groups differences. The sample size required to observe the same magnitude of mean difference (10%) with the same assumption of SD of 14 in the diabetes group after exercise training was 22. In the end we enrolled more (n=29 athletes and n=30 type 2 diabetes patients) because many started at the same time, we could not anticipate if/what number of drop-outs we would have and importantly, we had to complete the study within a certain time frame which was the shelf-life of the batch production isotopes for intravenous administration. There were not many drop-outs and we ended up with more participants than anticipated completing the whole study.

For the isotope analyses, we powered the study based on literature as we did not have our own data. We used the palmitate fractional synthesis rate reported by Perreault *et al* (*Obesity. (Silver. Spring)*).2010;18:1524-1531) of $0.23 \pm 0.04/h$ in pre-diabetes patients. The 0.04 in this paper is a SEM of between group differences, and their groups included both men and women (women have a much higher lipid turnover. We pre-specified a change of 20%, meaning a population mean difference =0.04, and expected SD for the difference to be 0.04 (we only included males for the specific reason to minimize variability and assessed within group changes), study power 0.8, significance 0.05. This required a sample size of n=10 in the diabetes group, and thus we matched the athletes' group, without prespecifying any athlete-diabetes differences. We ended up recruiting n=11 athletes and n=12 type 2 diabetes patients because the amount of isotopes we placed the order for with CK Isotopes resulted in a slightly higher number of bottles after portioning the sterile products and the clinical trails pharmacy using the necessary amounts required for pyrogenicity, particulate and stability testing, allowing us to do the additional 3 participants, or it would have been wasted.

We revised the statistical section by adding the detail of the sample sizes. We revised all Tables to show the mean (CI) of within group changes with each intervention, between group differences at baseline and post-intervention and, the between groups difference (CI) in their pre-to-post intervention changes. These have been mentioned in the statistical section, lines 520-546.

5. Results:

5a. Figure 1: Please add information on participant withdrawals to consort diagram.

The withdrawals (and the stage at which these occurred) have now been depicted on the Figure 1 – consort diagram.

5b. Table 1: Descriptive statistics of the study sample should be based on means (SD) and P-values removed as described above.

Done as requested.

5c. Pg. 5 line 137: Replace “Table 2” by “Table 1”.

5d. There seems to be a discrepancy in the reporting of baseline weight

between Tables 1 and 2.

Thank you for raising this, points 5c and 5d are addressed together here: The data presented in (now) Supplementary Table 1 refer to the assessment undertaken at the screening visit. During this visit, all participants were investigated thoroughly to eliminate any unknown subclinical conditions (for example valvular heart disease) or silent disease (for example approx. 30% of patients with Diabetes have silent myocardial infarctions that they are not aware). The prospective participants' level of activity/sedentarism was cross-checked objectively against their own perception declared through questionnaires, to make sure that they are either athletes or sedentary diabetes patients. All were all asked to perform a treadmill CPEX test to ensure that they are physically able to do this (some people think they can but are unable to do it in practice). Of note, the screening visit may have occurred in some participants weeks or even 1-2 months before the Study visit 1. This is because athletes wanted to schedule their deconditioning to fit around pre-planned races/competitions and type 2 diabetes patients needed to schedule their exercise training to ensure daily availability for attending to our unit for stationary bike exercise for 8 weeks out with any other planned activities (eg holidays, etc). The data presented in Supplemental Table 1 represents the full evaluation at screening of all those who eventually participated in the study (n=29 athletes, n=30 type 2 diabetes) – whereas all other tables show the data recorded at Study visit 1 (baseline) and Study visit 2 (post-exercise intervention). This is why the weight of participants may be a little different between screening visit and study baseline, as they will have been recorded at slightly different times. The data are correct as shown for the respective timepoints. We added “at screening” in the introductory Results paragraph.

REVIEWER COMMENTS

Reviewer #1 (Remarks to the Author):

No further comments

Reviewer #2 (Remarks to the Author):

1. The authors have made a reasonable effort to revise the paper according to the prior reviews. The paper now more fully captures the key data and interpretation. There are some remaining issues and concerns.
2. The authors should acknowledge the limitation of not having more direct measures of insulin sensitivity, which is a major limitation.
3. I disagree with the statement: The detrained athletes are the surrogate equivalent of matched healthy (untrained) volunteers for the type 2 diabetes patients. I still think that these post-intervention comparisons are not meaningful and should be omitted. A qualitative comparison of the trained vs detraining responses could be made, but it is too complicated to quantitatively compare these responses in such distinct subject groups.
4. How can the authors say that we observed that the saturated intramyocellular lipids (palmitate) turnover decreased proportionately much more than the linoleate post-deconditioning – when this was not significant? - trending statistical significance ($p=0.07$).
5. The authors still need to briefly discuss why specific IMCLs may be good non-exercise targets for intervention to improve health.

Reviewer #3 (Remarks to the Author):

Thank you for the improved revised manuscript. However, as noted in my first review, I strongly advise against reporting P-values in the table of baseline characteristics (Supplemental Table 1). Similarly, reporting of P-values in Supplemental Table 3, seems problematic because the study do not have a-priori hypothesis on diet.

**Subject: NCOMMS-23-12748-R2 - Intramyocellular Saturated Fatty Acid
Abundance and Turnover - an Index of Exercise-driven Metabolic Health**

REVIEWER COMMENTS

Reviewer #1 (Remarks to the Author):

No further comments

We would like to thank Reviewer #1 for their insightful and helpful peer review.

Reviewer #2 (Remarks to the Author):

1. The authors have made a reasonable effort to revise the paper according to the prior reviews. The paper now more fully captures the key data and interpretation. There are some remaining issues and concerns.

We would like to thank Reviewer #2 for their contribution to this manuscript through the peer review process.

2. The authors should acknowledge the limitation of not having more direct measures of insulin sensitivity, which is a major limitation.

We fully acknowledge the limitation of not having performed an oral glucose tolerance test for reporting on insulin sensitivity. This was for pragmatic reasons of avoiding a too prolonged fast and impractically long study visit (which would have become 9.5-10 hours), as explained in the previous point-by-point response. This has now been added at study limitations (page 14), apologies for the omission in Revision 1.

3. I disagree with the statement: The detrained athletes are the surrogate equivalent of matched healthy (untrained) volunteers for the type 2 diabetes patients. I still think that these post-intervention comparisons are not meaningful and should be omitted. A qualitative comparison of the trained vs detraining responses could be made, but it is too complicated to quantitatively compare these responses in such distinct subject groups.

Thank you for further clarifying, the Results section was re-arranged to report the numerical post-intervention values of abundance and turnover in the sections describing the "*Deconditioning in athletes*" and "*Exercise training in patients with Type2 Diabetes*". (The numerical values of intramyocellular abundance and turnover achieved after exercise interventions are not shown in Tables and these will be needed in the future if other researchers wish to either reproduce or compare our results with theirs). The section "*Post exercise intervention comparisons of athletes and patients with type 2 diabetes*" has now been completely removed from the manuscript. Similarly, all columns showing the "*Post exercise intervention comparisons of athletes and patients with type 2 diabetes*" and the corresponding column of *p values* have been removed from all Tables of results. The only "qualitative" post-exercise intervention left is the sentence referring to the blood lipid profile, which is an important clinical observation of the added effect of exercise on lipid-lowering therapy.

4. How can the authors say that we observed that the saturated intramyocellular lipids (palmitate) turnover decreased proportionately much more than the linoleate post-deconditioning – when this was not significant? - trending statistical significance (p=0.07).

Apologies for the over-representation of our explanatory effort in the previous point-by-point response, importantly, no such strong statement has been made in any of the versions of the manuscript.

5. The authors still need to briefly discuss why specific IMCLs may be good non-exercise targets for intervention to improve health.

We agree, the following paragraph has been added to the Discussion, pages 13-14:

Although we demonstrated significant beneficial changes in abundance and turnover of saturated intramyocellular lipids in response to exercise training in type 2 diabetes patients, physical exercise is not a practical solution for every individual. Further work should seek to explore which exercise mimetic pharmacotherapies are capable of recapitulating these findings in type 2 diabetes patients. In particular, solutions targeting insulin resistance through specific intramyocellular lipids or indirectly *via* upstream nutrient-sensing transcription factors ³⁴ targeting genes involved in specific lipid biosynthesis (such as long-chain fatty acid elongase 6 ³⁵ or Stearoyl-CoA desaturase ³⁶ respectively) could be explored.

Reviewer #3 (Remarks to the Author):

Thank you for the improved revised manuscript. However, as noted in my first review, I strongly advise against reporting P-values in the table of baseline characteristics (Supplemental Table 1). Similarly, reporting of P-values in Supplemental Table 3, seems problematic because the study do not have a-priori hypothesis on diet.

We thank Reviewer 3 for their input which has significantly improved the presentation of our manuscript. Apologies for the misunderstanding, have removed entirely the column of p values from Supplemental Table 1.

Similarly, we have now removed all the p value columns from Supplemental Table 3.